# Single-cell epigenomics and spatiotemporal transcriptomics reveal human cerebellar development

Suijuan Zhong[1,2,6,7] ✉, Mengdi Wang[3,4,7], Luwei Huang[3,4,7], Youqiao Chen[1,7], Yuxin Ge[1], Jiyao Zhang[1], Yingchao Shi[5], Hao Dong[3,4], Xin Zhou[1,2,6], Bosong Wang[1], Tian Lu[3,4], Xiaoxi Jing[1], Yufeng Lu[3], Junjing Zhang[1], Xiaoqun Wang [1,2,6] & Qian Wu [1,2,6] ✉

Human cerebellar development is orchestrated by molecular regulatory networks to achieve cytoarchitecture and coordinate motor and cognitive functions. Here, we combined single-cell transcriptomics, spatial transcriptomics and single cell chromatin accessibility states to systematically depict an integrative spatiotemporal landscape of human fetal cerebellar development. We revealed that combinations of transcription factors and cis-regulatory elements (CREs) play roles in governing progenitor differentiation and cell fate determination along trajectories in a hierarchical manner, providing a gene expression regulatory map of cell fate and spatial information for these cells. We also illustrated that granule cells located in different regions of the cerebellar cortex showed distinct molecular signatures regulated by different signals during development. Finally, we mapped single-nucleotide polymorphisms (SNPs) of disorders related to cerebellar dysfunction and discovered that several disorder-associated genes showed spatiotemporal and cell type-specific expression patterns only in humans, indicating the cellular basis and possible mechanisms of the pathogenesis of neuropsychiatric disorders.

The cerebellum is a central neural structure consisting of two hemispheres connected by the vermis. It is derived from the dorsal part of the most anterior hindbrain and located above the medulla oblongata[1]. The cerebellum is classically known to play a role in motor control, and it is now considered to contribute to many cognitive functions via connections with the cerebral cortex[2,3]. Injuries or diseases affecting the cerebellum have usually been linked to spinocerebellar ataxia, congenital cerebellar hypoplasia, intellectual disability, autism spectrum disorder (ASD), medulloblastoma, etc[4–8].

The actual cerebellum size expands faster than the neocortex throughout the evolution of apes[9]. The cerebellum, which only represents 10% of the brain mass, contains 80% of all brain neurons[10,11]. How large numbers of neurons are generated in the human cerebellum is a key question. Cerebellar neural progenitors are located within distinct niches and undergo programmed proliferation, differentiation and migration to eventually construct a functional cerebellum. The spatial organization of progenitors and neurons is essential for cerebellar development and function[12]. In the developing human

[1]State Key Laboratory of Cognitive Neuroscience and Learning, New Cornerstone Science Laboratory, Beijing Normal University, Beijing 100875, China. [2]IDG/ McGovern Institute for Brain Research, Beijing Normal University, Beijing 100875, China. [3]State Key Laboratory of Brain and Cognitive Science, Institute of Biophysics, Chinese Academy of Sciences, Beijing 100101, China. [4]University of Chinese Academy of Sciences, Beijing 100049, China. [5]Guangdong Institute of Intelligence Science and Technology, Guangdong 519031, China. [6]Changping Laboratory, Beijing 102206, China. [7]These authors contributed equally: Suijuan Zhong, Mengdi Wang, Luwei Huang, Youqiao Chen. ✉e-mail: zhongsuijuan@bnu.edu.cn; qianwu@bnu.edu.cn

cerebellum, rhombic lip (RL) progenitors and ventricular zone (VZ) progenitors give rise to glutamatergic (most are granule cells) and GABAergic neurons, respectively. After GABAergic neurons are generated, VZ progenitors are responsible for gliogenesis[13,14]. In the cerebella, arealization is also essential for neuronal connections and function. For example, different subtypes of Purkinje cells are located in different regions during development[15], and Purkinje cells are divided into different compartments with distinct neural circuits in the mature cerebella[15]. Recent studies have revealed the diversity and molecular features of human cerebellar progenitors and evolutionary differences in the cerebellum[12,16–20]. Although the molecular features of VZ and SVZ RL progenitors have been illustrated and linked to the origins of medulloblastoma[21,22], the global spatial gene expression pattern and epigenetic information at the single-cell level during human cerebellar development have not been comprehensively reported. Chromatin accessibility status as an epigenetic feature reveals the activity of cis-regulatory elements, which play essential roles in controlling spatiotemporal and cell type-specific gene expression patterns. The study of chromatin accessibility status not only enables us to reveal the regulatory programs of genes that govern human cerebellar development and cell differentiation but also might illustrate the evolutionary differences across species and help interpret how the mutations of noncoding DNA regions are linked to human diseases. Here, we combined single-cell chromatin accessibility states, spatial transcriptomes and single-cell transcriptomes to systematically depict an integrative spatial landscape of the molecular features and cellular composition of the developing human cerebellum covering GW 12-27. The multiomic data depict a comprehensive human fetal cerebellar spatial developmental blueprint that integrates the spatial information and dynamics of cell type diversity, as well as the molecular regulation of neuronal and glial cell fate determination at both transcriptomic and epigenetic levels, thus refining our knowledge of human cerebellar development and related disorders.

## Results

### Cell type diversity in the human fetal cerebellum

To investigate the cell type-specific gene expression patterns and regulatory dynamics during cerebellum development, we constructed a combination of chromatin accessibility profiles and gene expression from the entire left cerebellum at gestational weeks (GWs) 12-27 by single-cell ATAC sequencing and single-cell RNA sequencing to build a multiomic cell atlas of the developing human cerebellum (Supplementary Data 1–3; Fig. 1a, b and Supplementary Figs. 1–2). We classified 17 major cell clusters based on known markers: RL (rhombic lip) progenitors (*MKI67* and *ATOH1*), VZ (ventricular zone) progenitors (*PTF1A* and *MKI67*), granule cells (*MFAP4* and *MGP*), eCN (*NEFM* and *NEUROD1*), UBC (*EOMES*), Purkinje cells (*PCP4* and *RORA*), GABAergic neurons (*PAX2* and *GAD1*), Bergmann cells (*GDF10* and *PAX3*), astrocytes (*AQP4* and *SOX9*), OPCs (*OLIG1* and *PDGFRA*), oligodendrocytes (*MBP* and *APOD*), and microglia (*AIF1* and *C1QB*) (Fig. 1a, b, Supplementary Figs. 1c–e, 2c, d; Supplementary Data 4). We employed a peak calling approach, identifying a total of 527,539 open chromatin regions (OCRs), and 31,347 of these peaks overlapped with a protein-coding gene promoter. We profiled the pile-up of the ATAC-seq signal for each cluster with the top 200 cluster-enriched peaks to systematically confirm the cell identity determined using scATAC-seq. Classical transcription factor binding motifs were specifically detected in the corresponding clusters of cells, and Gene Ontology (GO) analysis showed that each cell type with specifically expressed peaks was involved in the corresponding biological processes (Supplementary Fig. 3a). Then, we analyzed the correlation between the cell type clusters identified from scATAC-seq and scRNA-seq and found that all scATAC-seq clusters were mapped to RNA-seq clusters (Supplementary Fig. 3).

### Spatial transcriptomics-defined arealization in cerebellar development

To characterize the regional gene expression profiles, we applied two methods of spatial transcriptome analyses, a slide-based technique (10x Genomics Visium) (GW12, 17) and a multiplexed-FISH-based technique (Transcription Factor sequential Fluorescence In Situ Hybridization, TF-seqFISH) (GW12, 19) (Fig. 1c–e and Supplementary Fig. 4). We defined the exact region information based on the human cerebellum anatomy structures[23] and validated the marker gene expression in the spatial transcriptomic results. Although the spot resolution in the Visium system was not at the single-cell level, it indicated 6007 and 3280 genes as the median number per spot at GW12 and GW17, respectively. In TF-seqFISH, we used probes for 1085 transcription factors (TFs) to illustrate the roles of TFs in cell fate determination. We integrated these two spatial datasets with the scRNA-seq data to map the spatial distributions and gene expression profiles of individual cells. We identified clusters of Visium spots with different transcriptional profiles that mapped to distinct locations (Fig. 1c, d, Supplementary Fig. 4d, e, i, j). Then, using the TF-seqFISH dataset as a bridge, we investigated the single-cell composition of each spot in Visium, which also indicated the spatial localization of all cells from scRNA-seq clusters (Fig. 1f, Supplementary Fig. 4f). The cell type localization and gene expression signatures were well characterized. For example, progenitors were located in both the VZ and RL at GW12; RORA+ Purkinje cells emerged early as the major type of neurons at GW12 and then at the outer edge of the cerebellar cortex at GW17; and AQP4+ astrocytes were detected only in the area close to the VZ in GW17, indicating that their generation from VZ progenitors occurred at a later stage than neurons (Fig. 1c–e, Supplementary Fig. 4).

We organized 21 gene modules at GW12 and found that these modules were highly correlated with subregions (Fig. 1g, h, Supplementary Fig. 5 and Supplementary Data 5). For example, genes from M19 are highly expressed in cells located in the PCL. GO analysis of these genes also indicated that they play roles in cerebellar Purkinje cell layer development and formation (Fig. 1i). Some modules are correlated with the cells located in several locations, such as M10, the genes of which are highly expressed in the RL, EGL and EGL migrating zones and are essential for neuron differentiation and regulation of synapse structure or activity. Additionally, we also found that genes belonging to the same GO term in some cases are involved in different modules. For example, genes from M10, M16, and M19 all play roles in neuron migration and are highly associated with subregions of the RL, EGL, EGL migrating zones and VZ progenitor differentiating zone, indicating that neuron migration may be regulated in these areas by using different sets of genes (Fig. 1g–i, Supplementary Fig. 5b).

### Molecular signatures of progenitors

We reconstructed developmental paths by RNA velocity without microglia, endothelial cells, T cells, Schwann cells, meninges or other neurons derived from outside the cerebellum to identify trajectories underlying neuronal and glial differentiation in the human fetal cerebellum (Fig. 2a, Supplementary Fig. 6a, b) using two paths with RL progenitors and VZ progenitors as start points. RL progenitors differentiated into excitatory neurons in the cerebellum, including granule cells, UBCs and eCNs, while VZ progenitors played roles in neurogenesis for GABAergic neurons and Purkinje cells and gliogenesis for astrocytes, OPCs, oligodendrocytes and Bergmann glial cells (Fig. 2a). By analyzing differentially expressed genes (DEGs) from scRNA-seq data, we identified essential genes expressed in these two types of progenitors. *NRN1* and *IGFBP5* were found to be alternative marker genes for RL progenitors, while *TTYH1* and *PTPRZ1* were considered to be alternative VZ progenitor markers (Fig. 2b; Supplementary Data 6). These regional progenitor markers were verified by analyzing spatial transcriptomic data and immunofluorescence staining (Fig. 2c and Supplementary Fig. 6c). The progenitors were

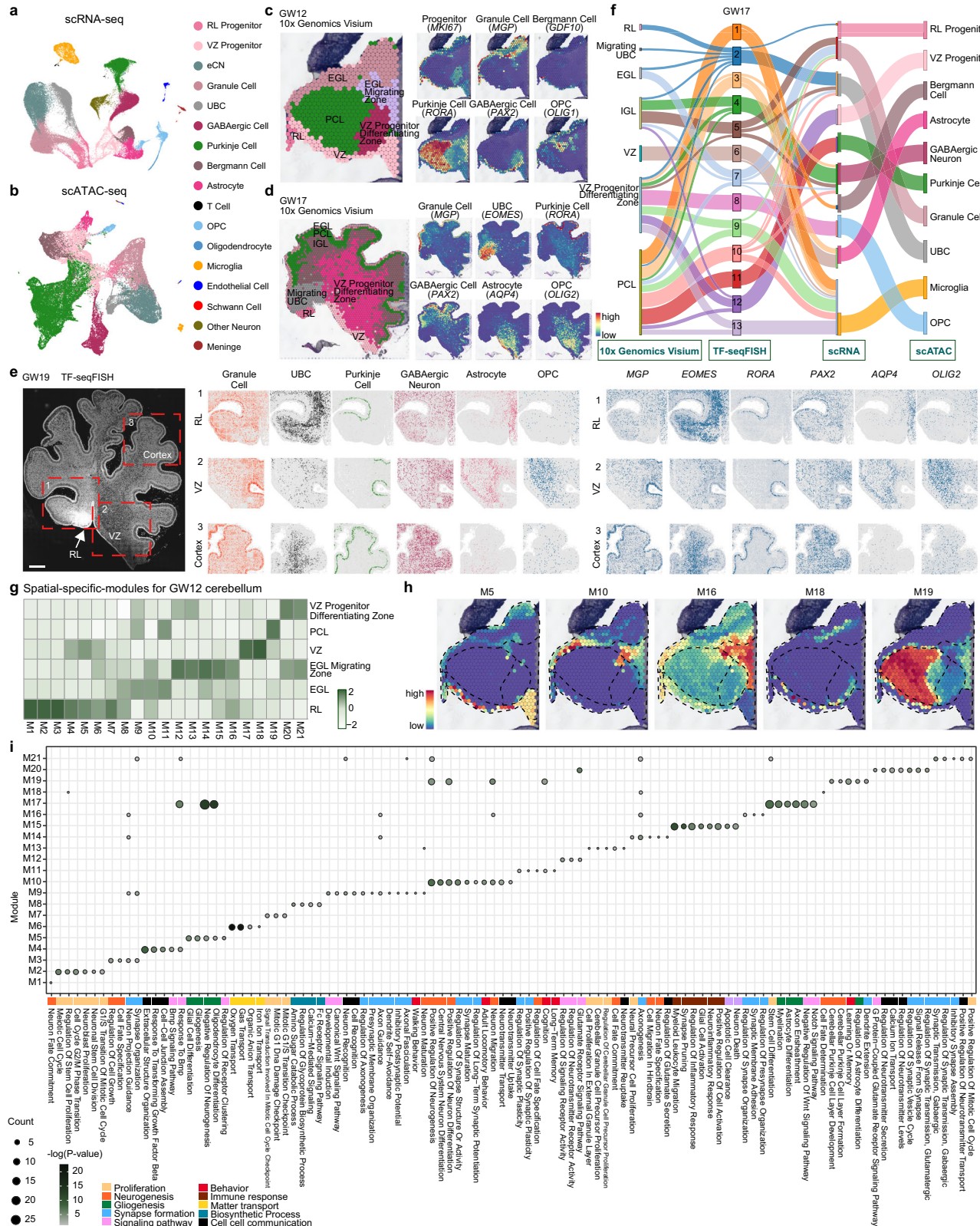

subclassified into 8 groups by the PCA algorithm (Progenitor 1-Progenitor 8) (Fig. 2d, Supplementary Fig. 6d, e; Supplementary Data 7). Combined with the trajectory analysis, we found that the subclusters of progenitors showed different cell differentiation potentials (Fig. 2e and Supplementary Fig. 6f). *OTX2*⁺ and *EOMES*⁺ progenitors were located in Clusters Progenitor 1 and Progenitor 2, suggesting that these cells were progenitors for UBC generation.

*STMN2* and *MGP* were expressed at high levels in Clusters Progenitor 3 and Progenitor 4, respectively, indicating eCN and granule cell fates. Cluster Progenitor 7 contained progenitors that expressed *LHX1* at high levels, indicating that these cells may differentiate into GABAergic neurons or Purkinje cells (Fig. 2e). Moreover, genes related to glial differentiation, such as *TTYH1*, *GFAP* and *FOXJ1*, were expressed at high levels in Progenitor 6 and Progenitor 8, respectively. Lineage analysis

**Fig. 1 | Temporal-spatial molecular diversity of single cells from the developing human cerebellum. a, b** Visualization of seventeen major classes of RNA-seq data (**a**) and thirteen classes of ATAC-seq data (**b**) in the developing human cerebellum using UMAP. Each dot represents a single cell, and cells are laid out to show similarities. Each cell color represents the cell type (RL, rhombic lip; VZ, ventricular zone; eCN, excitatory cerebellar nuclei cell; UBC, unipolar brush cell; OPC, oligodendrocyte precursor cell). **c, d** 10x Genomics Visium data showing the spatial distribution of different clusters in the GW12 (**c**) and GW17 (**d**) cerebellum. The expression of known markers is shown using the same layout on the right. **e** Images of DAPI staining showing the regions of TF-seqFISH in the GW19 cerebellum. TF-seqFISH plots showing the spatial distribution of different cell types in the GW19 cerebellum. Panels 1, 2, and 3 show the RL, VZ and cerebellar cortex, respectively. TF-seqFISH plots showing the gene patterns in the GW19 cerebellum using the same layout on the right. Scale bar, GW19, 500 μm. **f** Sankey plot showing the correlation between scRNA-seq, scATAC-seq, TF-seqFISH and 10x Genomics Visium Data in GW17. **g** Heatmap showing the spatial-specific modules for the GW12 cerebellum. **h** Gene patterns of each module shown in the same layout in (**c**). **i** Gene Ontology analysis of spatial-specific modules showing the KEGG pathways or biological processes in the GW12 cerebellum. Dots show the numbers of genes in each module, and the scale bar shows the -log(P-value) for the GO terms. Hypergeometric test.

indicated that Progenitor 6 could be the ancestor of astrocytes or Bergmann cells, while Progenitor 8 might differentiate into OPC and oligodendrocytes (Fig. 2d, e, Supplementary Fig. 6f–h).

## Cell fate determination by transcription factors

To investigate how gene expression in progenitors orchestrates and coordinates these cells to differentiate into different types of neuronal and glial cells, we conducted a regulon analysis based on scATAC-seq and scRNA-seq data. In detail, we identified the active TF regulons by their downstream gene expression and CREs detected via scATAC-seq. We first organized regulons into 20 modules and found that these modules were highly enriched in specific cell types (Supplementary Figs. 7a, b, and 8). Then, we depicted the network and trajectories of these regulons and built a landscape of the regulatory hierarchy by RL and VZ lineages (Fig. 2f, g). In the RL lineage, we observed that RL progenitors highly expressed specific TFs, such as *FOXM1* and *E2F2* (Fig. 2f, g, Supplementary Fig. 7c, Point 1). The GO analysis of E2F2 downstream genes indicated its role in cell cycle regulation (Supplementary Fig. 7d). The expression and promotor accessibility of *E2F2* decreased as developmental pseudotime proceeded (Fig. 2h–k). Next, we performed a coaccessibility analysis using Cicero and found that the regulatory network of *E2F2* varied in different cell types (Fig. 2l). E2F2, as a TF, showed distinct coaccessibility states in different cell types, indicating that E2F2 actively interacted with nearby distal elements in RL progenitors (Fig. 2l).

In the RL progenitor differentiation pathway, RL progenitors differentiate into neural cells driven by the expression of *KLF10* (Fig. 2g, Point 1), a cell cycle suppression-associated regulator[24]. Then, spatially restricted neural lineages were separated (Point 2), including the cerebellar cortex (*OLIG3* and *MSX1*) and the cerebellar nucleus (*FOXN3* and *MEIS3*) (Fig. 2g, m, n, Supplementary Fig. 7e). We then identified the regulated target genes of TFs at Point 2 and found that cerebellar cortex and nuclei development may be regulated by WNT signals and BDNF signals, respectively (Fig. 2o). The regulatory networks in the VZ lineage were more complex because neuronal and glial cells originated from the same region (Fig. 2 m, n).

VZ progenitors differentiated into neurons (*SOX4* and *TFAP2A*) and glial cells (*SOX2* and *HES5*; Point 4) (Fig. 2g). We further investigated the downstream genes of essential TFs at Point 4 and found essential genes involved in cell fate determination in two directions, GABAergic synapse and glial cell differentiation (Fig. 2g, p–r and Supplementary Fig. 7f–h). Collectively, our regulon map based on comprehensive analyses of scRNA-seq and scATAC-seq data suggests that TFs regulate cell fate in a hierarchical and combinatorial manner, and even the regulation of a single TF is dependent not only on its own RNA expression level but also on its regulatory capabilities, which exhibit cell type-specific and developmental timing patterns to guide and fine-tune human cerebellar development.

## The molecular and spatial characteristics of Purkinje cells

Purkinje cells, one type of GABAergic neuron, are responsible for conveying primary output from the cerebellar cortex to the CN, which controls the functions of movement and posture[25–27]. We performed

trajectory analysis of scATAC-seq data using Monocle3 in the developing cerebellum and built a path from VZ progenitors to Purkinje cells to examine the transcription networks regulating Purkinje cell development (Fig. 3a, Supplementary Fig. 9a, b). During the differentiation of VZ progenitors into Purkinje cells, a large number of the transcription start sites (TSSs) of highly expressed genes opened or closed in a pseudotime-dependent manner. For example, the TSS of *PTF1A*, an essential TF for VZ progenitor maintenance, closed, while the TSSs of several markers for Purkinje cells, such as *PCP4*, *RORA* and *RORB*, opened as pseudotime proceeded (Fig. 3b, c, Supplementary Fig. 9c; Supplementary Data 8). Interestingly, RORA and RORB are two forms of RAR-related orphan receptors, but they are expressed differently in the developing human cerebellum. *RORA* is known to be a classical marker of Purkinje cells[28], but *RORB* is not. The cells expressing these two genes were located distinctively, and RORB+ cells were only a subset of RORA+ cells (Fig. 3c and Supplementary Fig. 9d). By searching motif binding sites in the TSS, PTF1A, which is expressed at high levels by VZ progenitors, showed the potential to bind to the promoters of *RORA*, *RORB* and *CALB1*, indicating that PTF1A might play a role in regulating the expression of these downstream genes to guide Purkinje cell differentiation (Fig. 3c and Supplementary Fig. 9e; Supplementary Data 9).

We subclassified Purkinje cells into 9 groups using the PCA algorithm to further investigate the developmental characteristics of cerebellar Purkinje cells. The varying maturation status and DEGs in different subtypes of Purkinje cell also showed distinct spatial patterns (Fig. 3d–f, and Supplementary Fig. 9f). For example, C8 and C9 (*SOX2 + NFIA/NFIB +* ) cells were the precursors of Purkinje cells and located in VZ. Consistent with the spatial expression pattern, *RORB* was only expressed in three subgroups of Purkinje cells (Fig. 3d, f). The expression of RORA and RORB, as illustrated by RNAScope, also confirmed the scRNA-seq result that RORB⁺ Purkinje cells were a subset of RORA⁺ cells in the cerebellum at GW16 (Fig. 3g). By analyzing the scATAC-seq data for the Purkinje cells, we detected *RORB* motifs close to the TSSs of several classic marker genes of Purkinje cells, including *SCN2A*, *SLC12A5* and *DAB1* (Supplementary Fig. 9g). Furthermore, GO analysis of the target genes of *RORB* showed that *RORB* may play essential roles in Purkinje cell layer development, axonogenesis and synaptic signaling during cerebellar development (Fig. 3h). Cerebellar compartmentalization in stripes is a landmark structure of the cerebellar cortex. Accordingly, two subgroups were apparent from the spatial transcriptomic profiles with stripe-like compartmentalization (sp-P1 and sp-P2). The analysis of DEGs in these two Purkinje cell subgroups illustrated distinct molecular signatures (Supplementary Fig. 9h–k), indicating the spatial gene expression profiles of Purkinje strips during development.

## Developmental and spatial differences in granule cells

Granule cells are the most abundant excitatory neurons in the cerebellum. We first examined the gene expression and open status of chromatin in granule cell progenitors from the RL to investigate the developmental characteristics of granule cells and found that HEY1 is a key TF for determining granule cell fate (Figs. 2g and 4a–c;

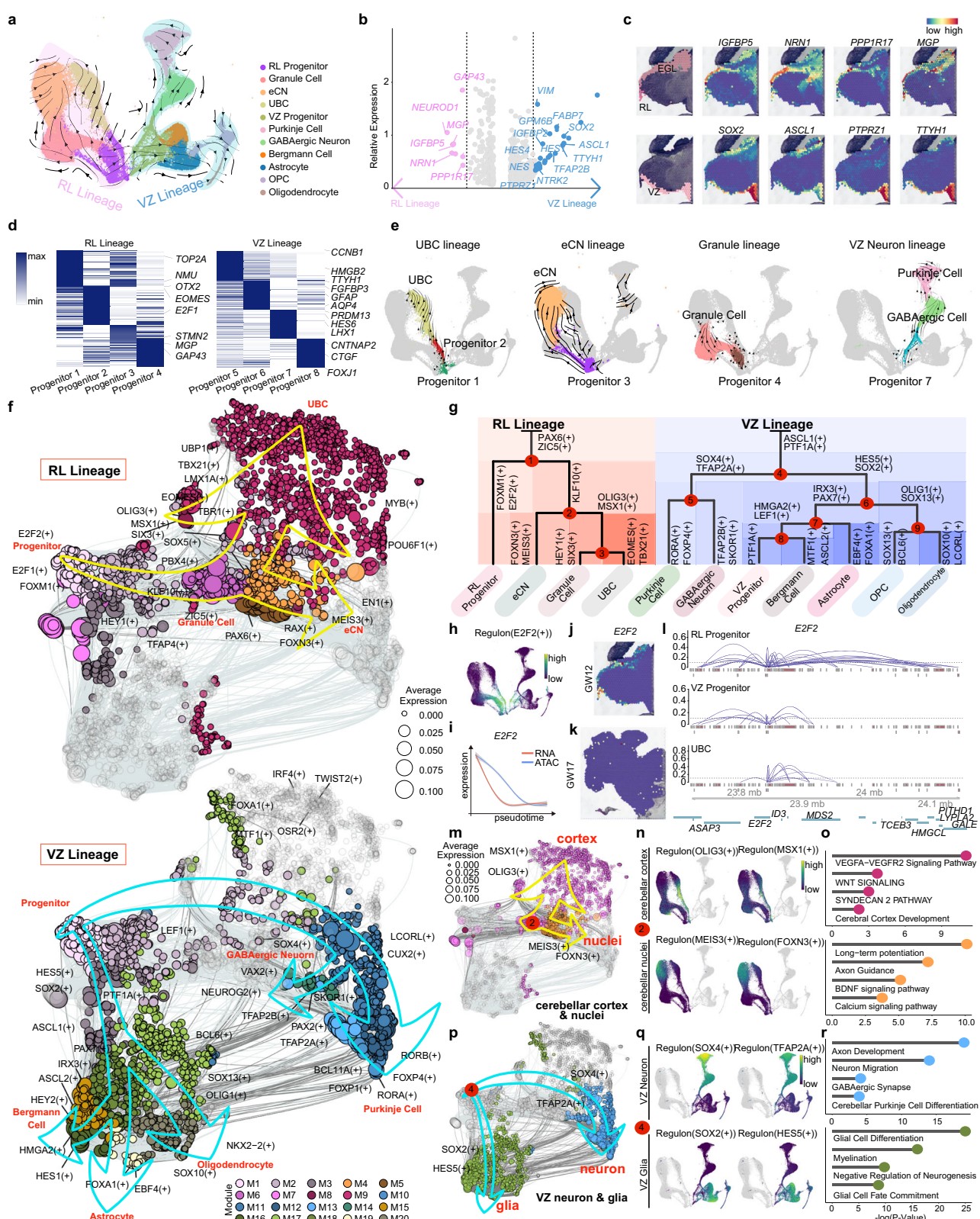

Supplementary Data 10).
 Furthermore, a GO analysis of HEY1 down-stream genes indicated that HEY1 plays roles in regulating granule cell precursor proliferation, neuronal migration and neuronal projection guidance (Fig. 4d).

We categorized granule cells into three groups according to their developmental stage to reveal their molecular characteristics, and GO terms for DEGs enriched in different stages indicated that cell division

was the major event occurring at GW12-14, followed by neuronal migration (GW16-21) and axonogenesis (GW24-27; Fig. 4e). Interestingly, the division and migration of neurons persisted from GW12-27, indicating that the granule cells continued to proliferate for a long time. We next investigated the differences between the early and late stages of cell proliferation in granule cells. We subgrouped all the granule cells and then analyzed the cell cycle and trajectory

**Fig. 2 | Cell diversity and regulon-typed cell-type specificity in the developing human cerebellum. a** Velocity visualization of the RL lineage and VZ lineage in the developing human cerebellum. **b** Scatterplot of all genes for correlation with the conserved differentiation network across the RL lineage (red plot) and VZ lineage (blue plot). **c** 10x Genomics Visium data showing the spatial distribution of RL and VZ in the GW12 cerebellum. The expression of region markers is shown using the same layout on the right. **d** Heatmap showing the expression level and identity of genes in RL progenitors and VZ progenitors. Specific gene expression in each type is shown on the right of the heatmap panel. **e** Velocity visualization of the UBC, eCN, granule cell and VZ neuron lineages using the same layout as in Fig. 2a. Cells from other lineages are colored gray. **f** Regulon-target modules showing the lineage trajectory in the developing cerebellum from progenitors to different cell types. RL lineage (top), VZ lineage (bottom). **g** A dendrogram of regulons on the top for each cell cluster in the main lineage in the cerebellum except for microglia, meninges, T cells, Schwann cells, endothelial cells and other neurons, showing the lineage trajectory in the developing cerebellum. The TFs shown at each branching of the dendrogram are representative of subjacent groups of regulons. **h** Regulon E2F2 patterns shown in the UMAP plots using the same layout as in Fig. 2a, black, no expression; yellow, relative expression. **i** Gene expression of RNA (red) and ATAC (blue) across pseudotime for E2F2. The shadow represents the 95% confidence interval around the fitted curve. **j, k** 10x Genomics Visium data showing the gene E2F2 spatial expression in the GW12 (**j**) and GW17 cerebellum (**k**). **l** Cicero coaccessibility in the region surrounding the E2F2 gene is shown for different cell types. **m** Regulon-target modules showing the differentiation point between the cerebellar cortex and nuclei in the RL lineage. **n** Feature Plots of Regulon MEIS3, FOXN3, OLIG3 and MSX1 in the RL lineage. **o** GO terms of the targets of the cortex or nuclei modules. Hypergeometric test. **p** Regulon-target modules showing the differentiation point between neurons and glia in the VZ lineage. **q** Feature Plots of Regulon SOX4, TFAP2A, SOX2 and HES5 in the VZ lineage. **r** GO terms of the targets of the VZ neuron or glial modules. Hypergeometric test.

(Supplementary Fig. 10a–d). We found that subgroups 2 and 5 were cells in S and G2/M phases and were located at the starting point of the trajectory map. We categorized these granule cells as early-stage and late-stage proliferating cells, and the DEGs indicated that NOTCH signals and WNT signals are involved in early and late stages, respectively (Fig. 4f, g; Supplementary Data 11).

Next, we examined granule cell migration at GW16, 21 and 27 using immunostaining. At GW16, only a few granule cells at EGL underwent inward migration, and IGL was observed at GW27 in humans (Fig. 4h). Since granule cells migrate and pass the Purkinje cells and Bergmann cells, which located between EGL and IGL, we next investigate cell–cell interaction between these cells by iTALK analysis. We found Purkinje cells and Bergmann cells both promote granule cells migration by the growth factor PTN signaling (Supplementary Fig. 10e–h). These observations indicated that the location of granule cells during development was very important for their biological processes. During granule cell generation, granule progenitors first differentiate and migrate from the RL to the EGL and then migrate from the EGL to the IGL. To understand the molecular features and regulatory signals of the whole process, we analyzed the spatial transcriptomes of GW12 and GW17. We analyzed 20 gene modules, and the genes of Module 2 were enriched in RL, playing roles in regulating the cell cycle (Fig. 4i). Genes of Module 6 were enriched in EGL at GW12, indicating that these cells were regulated by NOTCH signals (Fig. 4i–k, Supplementary Fig. 10i), which is consistent with the observation in early proliferating granule cells (Fig. 4g). The EGL at GW17 expressed different genes (Module 13) from the EGL at GW12, which were involved in the Hippo, MAPK, TGF-beta and BDNF signaling pathways (Fig. 4l–n, Supplementary Fig.10j). The IGL formed at GW17, and expressed genes involved in the regulation of cell migration and the WNT and BMP signaling pathways (Fig. 4n). Furthermore, we analyzed the spatial transcriptome data and divided them into 4 subgroups (G1-G4), which exhibited distinct regionalization in GW17 tissues. Cells of G1 were considered to be EGLs, while G2-G4 represented IGLs in different lobules with distinct gene expression characteristics (Fig. 4o). Collectively, we found that Purkinje cells and granule cells showed temporal-spatial transcriptional differences and may interact and develop in a coordinated manner.

### Species differences in mammalian cerebellar development
Compared to IGL formation in humans during prenatal development, IGL formation in mice occurs postnatally[29,30], indicating that the development of the cerebellum may exhibit evolutionary divergence between rodents and humans. Hence, we used the Glmnet algorithm[31,32] to age match the developing cerebellum between mice and humans based on single-cell transcriptome profiles. We found that the human GW12-GW27 cerebellum was comparable to that of mice from embryonic Day 15 (E15) to postnatal Day 2 (P2) (Fig. 5a and Supplementary Fig. 11a–c), indicating that the development of the cerebellum

is initiated earlier in humans. We next compared scRNA-seq data from human (GW12-27) and mouse (E15-P2) cerebella[33] and analyzed the differences in cell type and gene expression profiles between the species using the CCA algorithm (Fig. 5b, c; Supplementary Data 12, 13). The majority of cell types were similar across the species but still showed different gene expression profiles (Fig. 5c). Intriguingly, we also found that several subtypes of neurons showed human specificity, including eCN (Cluster 49), granule cells (Cluster 50), Purkinje cells (Cluster 51), and RL progenitors (Cluster 45; Fig. 5b).

Notably, by comparing human and mouse Purkinje cells, we found 246 genes that were selectively expressed in humans (Fig. 5d). An analysis of spatial information for these genes also indicated the regional expression of these genes, such as *RORB*, *CA8*, *FOXP1* and *CNTNAP2*, which were specifically expressed at high levels in human Purkinje cells (Fig. 5d–f). As described above, RORB is a marker gene we observed to label subgroups of Purkinje cells in humans (Figs. 3d–g, 5e, f) that may play an essential role in regulating Purkinje cell differentiation and maturation. Immunostaining further validated that RORB was detectable in human Purkinje cells at GW16 and GW27 (CALB1+ cells) but not in the mouse Purkinje layer at P2 (Fig. 5g); this was not due to the species specificity of the antibody, since we observed Rorb+ cells in the mouse cerebral cortex from the same sample (Supplementary Fig. 11d). Interestingly, based on in situ results from developing and adult mouse brains (www.brain-map.org), we found that *Rorb*+ cells exist in mouse VZ progenitors at E13.5 and E15.5 but not in differentiated Purkinje cells (Supplementary Fig. 11e–k), indicating that Rorb may play a role in mouse Purkinje cell development only at an early stage.

Compared to the other progenitors, human-specific RL progenitors of Cluster 45 differentially expressed genes localized in the RL and EGL, indicating the cell fate to granule cells (Fig. 5h, i). These RL progenitors exhibited high expression of *ARHGAP11B*, a human-specific gene that is considered to be involved in cerebral cortex expansion and folding in mammals[34–36] (Fig. 5h, i). We transiently expressed *ARHGAP11B* in mouse cerebellar progenitors at E11.5 to further investigate the role of *ARHGAP11B* in cerebellar development and observed cerebellar cortex expansion and extra folded structures at P12, which may result from an increase in the number of granule cells (Fig. 5j, k, Supplementary Fig. 11l–n) by mitochondria-based progenitor metabolism regulation[37]. Collectively, we found that although the cerebellum has been considered to be an evolutionarily conserved region of the brain, the human cerebellum still exhibits differences in cell type and gene expression profiles compared to mice during development.

### Disease-related genes and SNPs in different cell types
Mutations in coding and noncoding regions are associated with disorders and diseases which cause cerebellar dysfunction. Next, we analyzed the expression and chromatin accessibility patterns of disease-associated genes as well as single-nucleotide polymorphisms

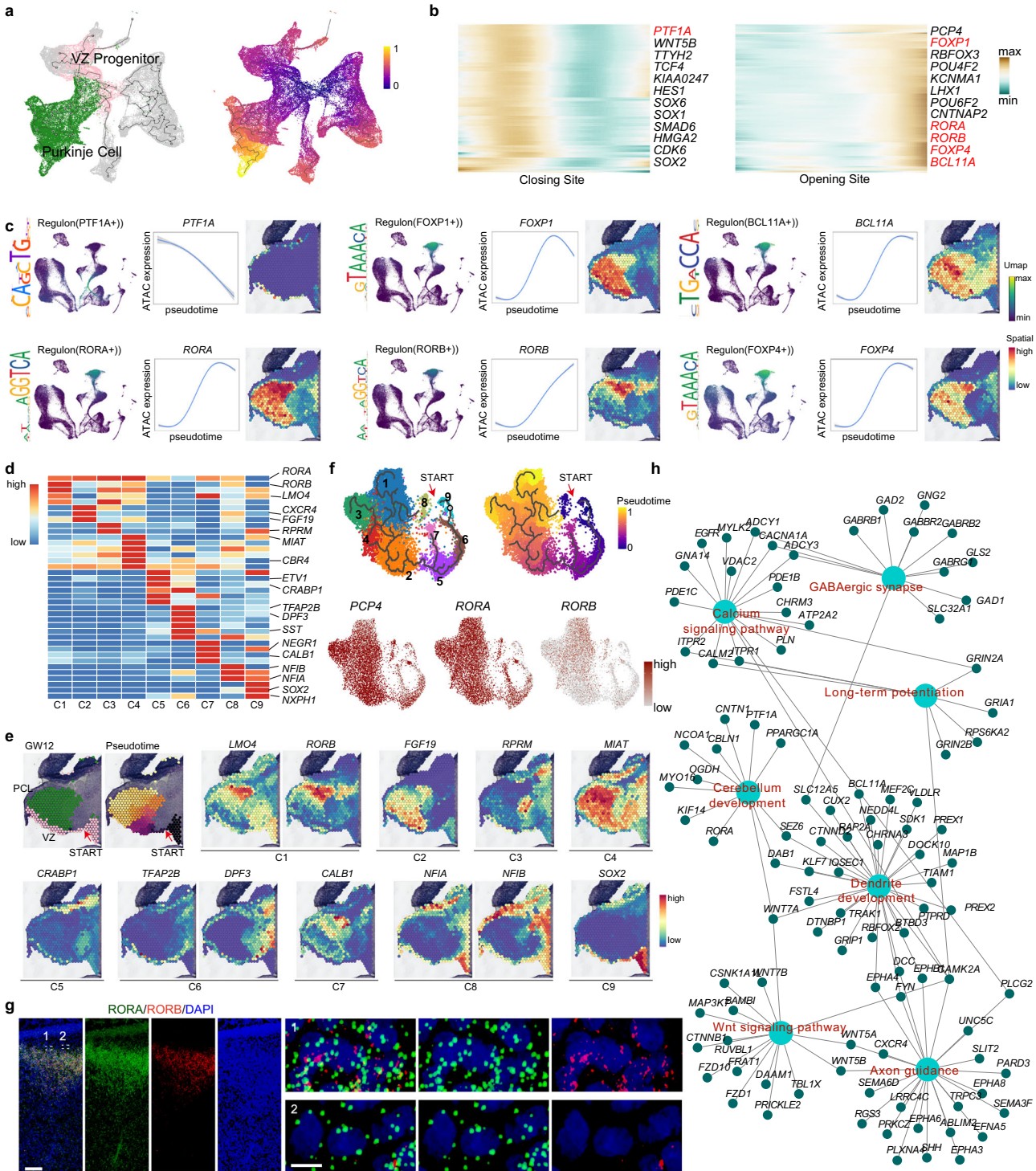

**Fig. 3 | Regulatory networks of Purkinje cell development. a** Cell lineage relationships of progenitors and Purkinje cells analyzed in the developing human cerebellum in ATAC-seq using the same layout as in Fig. 1b (ATAC). Monocle recovered a branched single-cell trajectory beginning with progenitors and terminating at Purkinje cells. Each dot represents a single cell; the color of each cell represents the cell type (left) and pseudotime (right). Cells from other lineages are colored gray. **b** Smoothed pseudotime-dependent accessibility curves of VZ progenitor and Purkinje cells generated by negative binomial regression and scaled as a percent of the maximum accessibility of each site. The top 10000 highly expressed sites with pseudotime-dependent accessibility are shown. **c** Regulon PTF1A, FOXP1, BCL11B, RORA, RORB and FOXP4 patterns shown in the UMAP plots using the same layout as in Fig. 1b (RNA), black, no expression; yellow, relative expression. The graphs on the left show the motif sequences of the regulon (left); gene expression of ATAC (blue) across

pseudotime for the above genes (middle), the shadow represents the 95% confidence interval around the fitted curve; 10x Genomics Visium data showing the gene spatial expression in the GW12 cerebellum. **d** Heatmap showing the expression level and identity of marker genes in Purkinje cell subtypes. **e** The pseudo-time and subtype-specific gene patterns in GW12 spatial data. **f** Trajectory analysis of nine subtypes of Purkinje cells in the developing human cerebellum using UMAP and FeuturePlots showing *PCP4, RORA* and *RORB* expression. Each dot represents a single cell, and cells are laid out to show similarities. Each cell color represents the cell type. **g** RNAscope images of RORA and RORB in the GW16 cerebellum. Scale bar, 500 μm (top), 100 μm (bottom). The experiments were repeated three times independently with similar results. **h** Gene ontology analysis of target genes of Regulon RORB showing the predicted function of target genes of Regulon RORB in Purkinje cells in the developing human cerebellum.

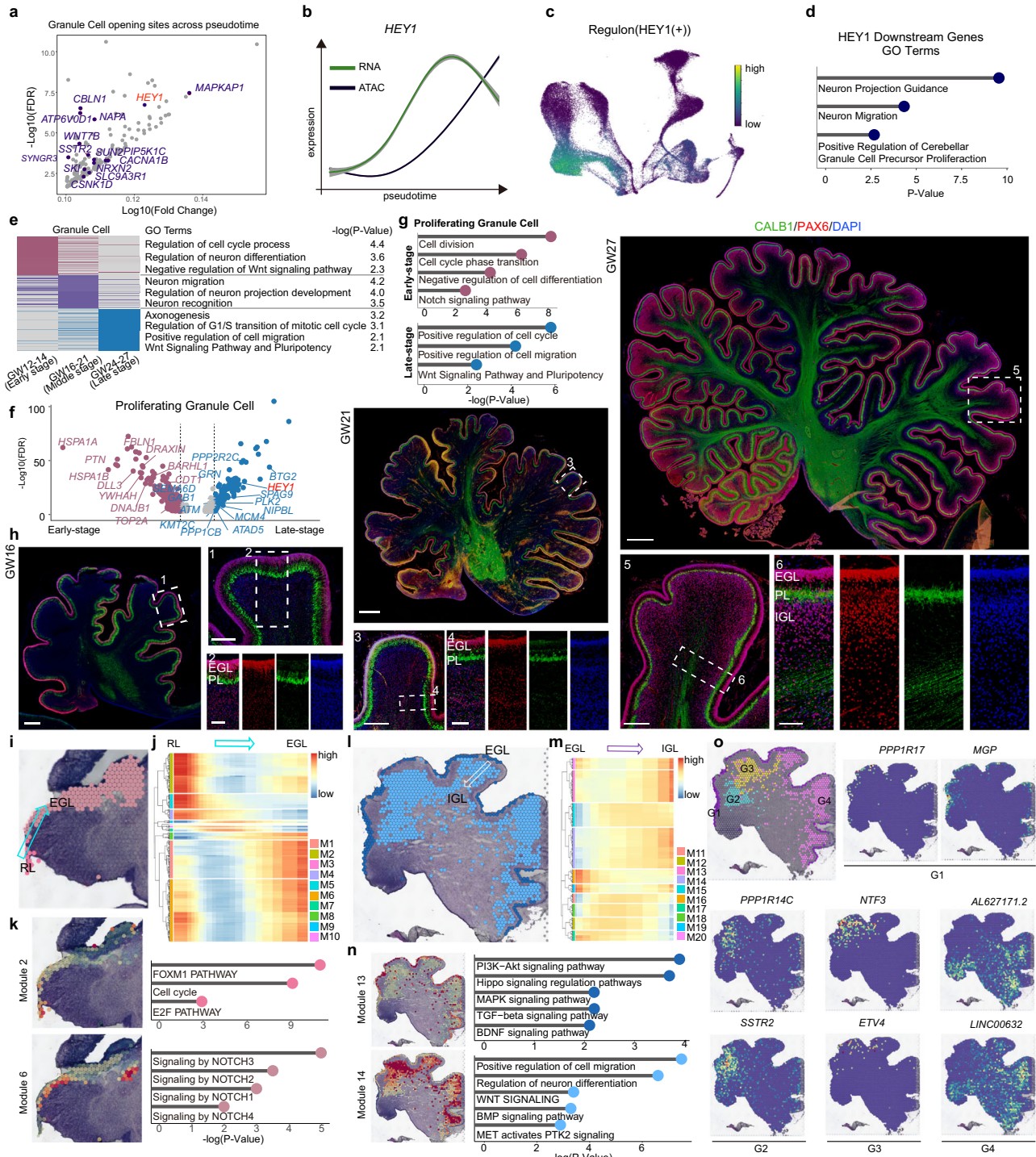

**Fig. 4 | Dynamics of neurogenesis of granule cells in the developing human cerebellum. a** Scatter plots depicting highly expressed peaks annotated by mapping to the hg19 human genome in granule cells. **b** Gene expression of RNA (green) and ATAC (black) across pseudotime for HEY1, the shadow represents the 95% confidence interval around the fitted curve. **c** Regulon HEY1 patterns shown in the UMAP plots using the same layout as Fig. 1b (RNA), black, no expression; yellow, relative expression. **d** GO terms of the target genes of regulon HEY1. Hypergeometric test. **e** Heatmap showing the differences in developing stages of granule cells. Gene ontology analysis showing the biological functions of different stages of granule cells (right). Hypergeometric test. **f, g** Scatter plots (**f**) and gene ontology (**g**) analysis showing the differentially expressed genes between early- (red plot) and late-stage (blue plot) proliferating granule cells. Hypergeometric test. **h** Immunofluorescence images of CALB1 and PAX6 in GW16, GW21 and GW27. Scale bar, GW16, 500 μm (left), 200 μm (right, top), 100 μm (right, bottom); GW21, 1000 μm (top), 300 μm (left, bottom), 100 μm (right, bottom). GW27, 1000 μm (top), 300 μm (left, bottom), 100 μm (right, bottom). The experiments were repeated three times independently with similar results. **i–k** Spatial-specific gene modules showing the gene patterns from the RL to EGL differentiation process (**i, j**). Modules 2 and 6 are related to the RL and EGL regions, respectively. GO terms showing the KEGG pathways in this process (**k**). Hypergeometric test. **l–n** Spatial-specific gene modules showing the gene patterns from the EGL to IGL differentiation process (**l, m**). Modules 13 and 14 were related to the EGL and IGL regions, respectively. GO terms showing the KEGG pathways in this process (**n**). Hypergeometric test. **o** 10x Genomics Visium data showing the spatial distribution of four subclusters of granule cells and the expression of different markers in the GW17 cerebellum.

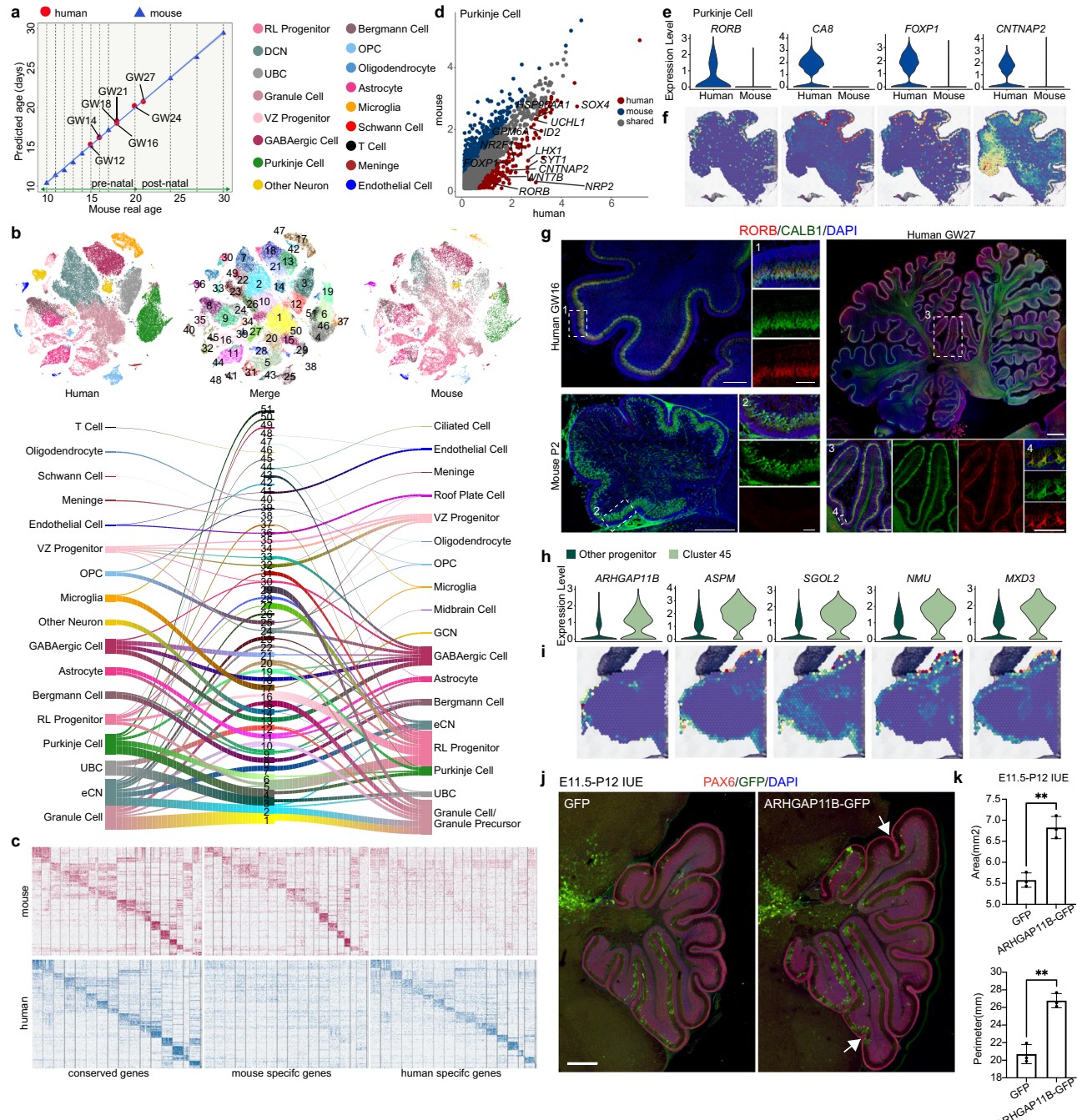

**Fig. 5 | Specific genes expressed in the developing human cerebellum. a** The chart shows the predicted mouse age of the developing human cerebellum. The blue triangle shows the real age and predicted age of the developing mouse cerebellum. The red dot shows the predicted mouse age of human data (GW12: E15, GW14: E16, GW16-20: E18, GW24: P0, GW27: P2). **b** tSNE visualization of human (GW12-27) and mouse (E15-P4) neural lineage cell types analyzed using CCA and color-coded based on CCA joint clusters (top middle) and cell type. The top left graph shows human cell types, and the top right graph shows mouse cell types, with each dot representing a single cell. River plots comparing cell type assignments for humans and mice with CCA joint clusters. **c** Heatmap showing the conserved genes, mouse-specific genes and human-specific genes between the human (GW12-27) and mouse cerebellum (E15-P4). **d** Pairwise comparison of gene expression in Purkinje cells between humans and mice. The red dots (human) and blue dots (mouse) indicate differentially expressed genes across species. **e** Vlnplots of *RORB, CA8,*

*FOXP1* and *CNTNAP2* expression in Purkinje cells between humans and mice. **f** 10x Genomics Visium data showing *RORB, CA8, FOXP1* and *CNTNAP2* expression in the GW17 cerebellum. **g** Immunofluorescence images of RORB and CALB1 in human GW16 and mouse P0. Scale bar, 300 μm (left) and 100 μm (right) for human GW16; 300 μm (left) and 100 μm (right) for mouse P2. The experiments were repeated three times independently with similar results. **h** Vln plots of *ARHGAP11B, ASPM, SGOL2, NMU* and *MXD3* expression in progenitors in the human cerebellum. **i** 10x Genomics Visium data showing *ARHGAP11B, ASPM, SGOL2, NMU* and *MXD3* expression in the GW12 cerebellum. **j** Overexpression of ARHGAP11B at E11.5 promotes cerebellar cortex folding observed at P2 in mice. Scale bars, 500 μm. **k** Quantification of the area and perimeter of the mouse cerebellar cortex at P12. **P < 0.01, P value(area)=0.0024, P value(perimeter)=0.0015, two-sided t-test, *n* = 3 biologically independent animals. mean ± s.e.m.

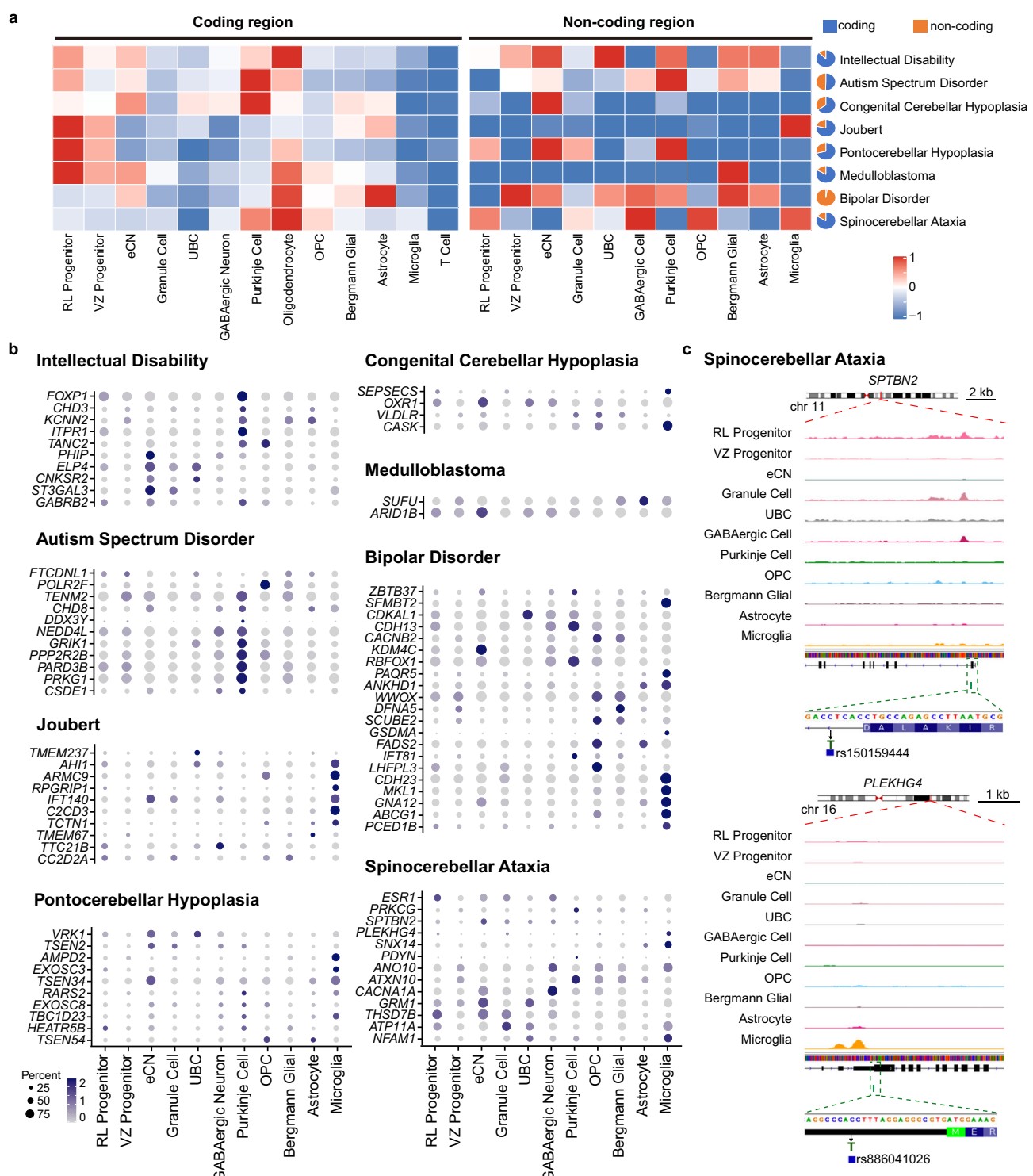

**Fig. 6 | Diseases in the human cerebellum. a** Aggregated expression of disease-associated genes in each cell type (coding region, left; noncoding region, right). **b** Expression patterns of selected disease-associated genes in each cell type in the ATAC dataset. **c** Normalized ATAC-seq profiles of *SPTBN2* and *PLEKHG4* in the cerebellum, with all cell types showing activation of *SPTBN2* and *PLEKHG4*.

(SNPs) in different cell types, including spinocerebellar ataxia, congenital cerebellar hypoplasia, pontocerebellar hypoplasia, intellectual disability, Joubert syndrome, autism spectrum disorder (ASD), medulloblastoma and bipolar disorder (Fig. 6a, b, Supplementary Fig. 12; Supplementary Data 14). In general, 97.5% of bipolar disorder-associated SNPs and 49.5% of ASD-associated SNPs were located in noncoding regions, while in the other disorders, the majority of SNPs were located in coding regions (Fig. 6a). We found that coding genes

related to pontocerebellar hypoplasia and Joubert syndrome, two developmental diseases, were enriched in RL and VZ progenitors. SNPs in coding and noncoding regions involved in ASD were both enriched in Purkinje cells. We further analyzed the cell type specificity of each gene categorized by disorder using scRNA-seq data for SNPs in the coding regions (Supplementary Fig. 12a) and scATAC-seq data for SNPs in the non-coding regions (Fig. 6b), respectively. Among these 1,619 genes with SNPs, we found that 1,355 genes show SNPs either in the

coding regions or non-coding regions, while 264 genes contain SNPs both in coding and non-coding regions. Among these 264 genes, 46.59% of genes have the SNPs that are associated with different disease phenotypes (Supplementary Data 15). For example, the *CNTNAP2* gene, which was identified as a gene responsible for speech-language delay in individuals with ASD[38], was expressed at high levels in Purkinje cells. We next analyzed the open state of chromatin in noncoding regions where the disease-related DNA mutations are located and the cell-type specificity (Fig. 6b, c, Supplementary Fig.12b). The open state of chromatin of the *FOXP1* gene, dysfunction of which has been reported to be associated with intellectual disability and speech defects[39,40], was enriched in Purkinje cells. Notably, *CNTNAP2* and *FOXP1* showed species specificity in developing human Purkinje cells (Fig. 5e, f). In addition, a C-to-T substitution in the *SPTBN2* and *PLEKHG4* (puratrophin-1) genes is known to be associated with spinocerebellar ataxia[41–43]. We found that the chromatin at the mutation location was open in RL progenitors, granule cells and GABAergic cells for *SPTBN2* and only in microglia for *PLEKHG4* in the developing human cerebellum, indicating that cerebellar disorders may be associated with not only cell type-specific gene expression profiles but also chromatin accessibility states (Fig. 6b, c).

## Discussion

By performing high-throughput parallel single-cell profiling of chromatin states and gene expression, we constructed a comprehensive spatial gene expression and regulatory atlas of the early- and mid-gestational human cerebellum, revealing the molecular characteristics and gene expression dynamics of multiple cell types during this developmental process. Notably, these multiomic data also illustrate the molecular regulation governing progenitor differentiation and cell fate determination during neurogenesis and gliogenesis. Our network indicates that cell fate determination occurs in a hierarchical manner and that a combination of several TFs is essential for each point of cell fate divergence. Interestingly, both the dynamic expression of a TF gene in different cell types at different developmental times and the coaccessibility of a TF show cell type and temporal differences, which is similar to human cortical development[44], indicating that the intricate gene expression regulated by TFs during human cerebellar development is tightly associated with chromatin structure status.

We combined the single-cell transcriptomic profiles with spatial information, adding valuable information about the underlying developmental programs in the cerebellum. Previous studies have described the spatial transcriptomic information of RL and VZ progenitors but not differentiated cells at single-cell resolution[12]. Purkinje cells and granule cells are known to exhibit spatial patterns with distinct molecular signatures, but these studies are usually based on 10x Genomics Visium data and tissue immunostaining[15,45]. Hence, our integrated spatial-multiomics landscape helped us to link gene-coded cell type and structural development to circuit formation and function diversification in different areas of the human cerebellar cortex[2]. In the early developing cerebellum (GW12), FOXP1+ and RORB+ cells could be considered to be two subgroups of Purkinje cells located in different regions. We also found the specific gene expression pattern of Purkinje cells in different sublocations at early development stages. Purkinje cells located in different stripes have distinct neural projections and connectivities, and these differences might be initiated in early developing Purkinje cells.

The human cerebellum develops over a long time, from 30 days postconception to 2 years postnatally[46]. Based on single-cell transcriptome similarity, our analyses indicate that the developmental stages of E15.5 to P2 mice are similar to the human developmental stages of GW12-27, indicating that the human cerebellum develops much earlier than the mouse cerebellum; however, the overall cell type emerging sequence is conserved. In addition to its important role in motor control, the human cerebellum is also involved in a wide variety

of cognitive functions[47,48]. Cross-species comparisons at the single-cell transcriptomic level indicate that some cell subtypes are only present in humans. Since we analyzed a developmental dataset, the subtype of cells may only reflect the transient state of the cell during development. However, we still propose that some of these results may provide an opportunity to understand the evolution and function of the human cerebellum. First, we identified *RORB* as a marker gene for some subgroups of developing Purkinje cells, which may play a role in regulating Purkinje cell differentiation and maturation. RORA and RORB are both retinoic acid-related orphan receptors and shared relatively high sequence identity and conserved domains[49]. *RORA* is a classic marker of Purkinje cells and RORB+ Purkinje cells are only a subset of RORA+ cells, showing distinct regionalization in early development but a relatively overlapping distribution later. Notably, unlike *RORA*, which is an evolutionarily conserved gene, *RORB* is not expressed in Purkinje cells but only in VZ progenitors in mice. *RORB* is expressed at high levels in the brain and retina, serving as a conservative cortical layer IV neuronal marker[50,51] and playing a critical role in retinal neurogenesis. According to a recent study, a loss-of-function mutation in RORB disrupts saltatorial locomotion, resulting from defects in the differentiation of populations of spinal cord interneurons in rabbits[52]. However, the function of RORB in human Purkinje cells requires further investigation.

Granule cells are the most abundant neuronal type in the adult human brain[29,53]. The human cerebellar cortex evolved not only in terms of the remarkable expansion of the surface with more complex folded structures but also in the total granule cell number. Cerebellar granule cell precursors are generated from RL progenitors and migrate to form the EGL, which then give rise to granule cells and migrate to the IGL. Developmental data are important because they describe the molecular features of progenitors, which may provide some clues about the mechanisms of cerebellar cortex expansion. Granule cells from all developing stages (GW12-27) are distributed evenly in subgroups, indicating that the granule cells experienced a long time of proliferating and differentiation, which could be how to make a huge population at the end. Intriguingly, we found that the human-specific gene *ARHGAP11B*, which is expressed in human cortical basal progenitors and promotes human cerebral cortex expansion and folding during evolution[34–36], was also expressed at high levels in RL progenitors, distinguishing these cells from mouse RL progenitors. The expression of ARHGAP11B in mouse cerebellar progenitors leads to mouse cerebellar cortex expansion with more granule cells and more complex folded structures. During human cortical development, ARHGAP11B promotes the proliferation of basal progenitors by inhibiting the mitochondrial permeability transition and inducing glutaminolysis[37]. Our observation suggests that the expansion and folding of the human cerebellar cortex occurs in concert with that of the human neocortex and possibly shares similar molecular mechanisms and gene evolution. In addition, we observed that granule cells were continuously generated from early to mid-gestational stages and observed a subset of granule cells that are dominant in humans, characterized by genes typically expressed by mature neurons, indicating that the generation and maturation of granule cells occurs earlier in humans than in mice during fetal cerebellar development. This feature of early initiation and long-lasting genesis is likely responsible for granule cells being the most abundant neuronal type in the brain[53].

Although our spatial multiomic dataset only covered early to mid-term development of the human fetal cerebellum, this period is a key window for neurogenesis and gliogenesis, which are fundamental for the structure and function of the cerebellum in adults. In addition to illustrating the landscape of human cerebellar development, our studies serve as a comprehensive dataset that will facilitate further research on the effect of cerebellum-related disorders, including neurodevelopmental and adult-onset disorders. Previous studies

indicate that RL progenitor cells located at the subventricular zone are the source of Group 3 and Group 4 medulloblastoma[22]. Interestingly, we found that *FOXP1* and *CNTNAP2*, two disorder-associated genes[38–40], were not only highly expressed in Purkinje cells during development but also specifically highly expressed in humans but not mice, indicating that these disorders may be caused by abnormal development, especially dysfunction of Purkinje cells in humans. Intersection analyses of GWAS data and our single-cell multiomic data will help not only in the investigation of cell types as origins of diseases but also provide genome mutation maps either at the coding region or at the transiently accessible loci with cell type and developmental timing information, which will be a very powerful tool for studying the pathogenesis of neuropsychiatric disorders.

In summary, we combined single-cell chromatin accessibility states, spatial transcriptomes and single-cell transcriptomes to systematically depict an integrated spatial landscape of the molecular features and cellular composition of the developing human cerebellum covering GW12-27. We revealed that the cell fate determination was in a hierarchical manner and a combination of several TFs is essential for each point of cell fate divergency, offering a very informative TF map and their accessibility states for cell fate determination. As a result of spatial transcriptome analysis, we found that not only progenitor cells at different locations (such as RL and VZ) showed differential gene expression profiles, but also developing neurons (such as Purkinje cells and granule cells) displayed spatial-temporal distinctive molecular signatures, which may be associated with their function. The cross-species comparison between human and mice indicated that some cell subtypes and genes were only presented in humans. Firstly, we found RORB as a marker for some subgroups of developing Purkinje cells, which played roles in regulating Purkinje cell differentiation and maturation. Notably, RORB was not expressed in Purkinje cells, but only in VZ progenitors in mice. Secondly, we also found the human-specific gene ARHGAP11B that were reported to promote human cerebral cortex expansion and folding in evolution, also highly expressed in RL progenitors. Intriguingly, expressing ARHGAP11B in mouse cerebellar progenitors led to increase of granule cells and cerebellar cortex expansion, as well as more complex folding structures, suggesting that the expansion and folding of the human cerebellar cortex occurs in concert with that of the human neocortex and possibly shares similar molecular mechanisms and gene evolution. Finally, we also revealed that SNPs associated with cerebellar developmental disorders could be mapped to coding and noncoding regions in specific cell types. We discovered that several disorder-associated genes showed spatiotemporal and cell type-specific expression patterns only in humans, suggesting the possible pathogenesis mechanisms of human neuropsychiatric disorders.

## Methods

### Human subjects
The de-identified human tissue collection and research protocols were approved by the Reproductive Study Ethics Committee of Beijing Anzhen Hospital (2014012X) and the institutional review board (ethics committee) of the Institute of Biophysics (H-W-20170110). Beijing Anzhen Hospital was in charge of recruiting donors for this research. The researchers explained the research purpose and the study plan, and the printed informed consent documents were provided to the patients. The fetal tissue samples were collected after the donor patients signing informed consent document. No participant compensation was provided. The sample number of fetal tissue was 15 samples for scRNA-seq, 9 samples for scATAC-seq, and 12 samples for spatial transcriptome experiments.

### Mice
Animal housing and all experimental procedures in this study were in compliance with the guidelines of the Institutional Animal Care and

Use Committee of the Beijing Normal University (IACUC(BNU)-NKLCNL2019-09). All mice had free access to food and water, were housed in the institutional animal care facility with a 12 h light-dark schedule. The humidity was kept at 50-65%. Rooms and cages were kept at a temperature range of 20-26 °C. All the subjects were not involved in any previous procedures. WT CD-1 mice purchased from Charles River Laboratories in China (Vital river, Beijing, China).

### Tissue sample collection and dissection
We assigned the embryonic tissues age using gestational age measured in weeks from the first day of the woman's last menstrual cycle to the sample collecting date. Fetal cerebellum was collected in ice cold artificial cerebrospinal fluid containing 125.0 mM NaCl, 26.0 mM NaHCO$_3$, 2.5 mM KCl, 2.0 mM CaCl$_2$, 1.0 mM MgCl$_2$, 1.25 mM NaH$_2$PO$_4$ at a pH of 7.4 when oxygenated (95% O$_2$ and 5% CO$_2$). We put the cerebellum in hibernate E medium (Invitrogen, Cat.A1247601) and washed for three times, then dissected in fresh hibernate E medium.

### RNA Library preparation for high-throughput sequencing
We aimed to collect whole cells for RNA library preparation. The cerebellum tissue was first digested in 2 mg/ml collagenase IV (Gibco, Cat.17104-019) and 10 U/µl DNase I (NEB, Cat.M0303L) in hibernate E medium and then 1 mg/ml papain (Sigma, Cat.P4762) and 10 U/µl DNase I in hibernate E medium. We vortexed and kept the tissue in 37 °C on thermo cycler with 300 g for 20 min. Further pipet was adopted to fully digest the tissue into single cells. After that, the cell suspension was centrifuged at 500 g for 5 min to get the cell pellet. The digestion medium was carefully removed and the cell pellet was resuspended in 300 µl 0.04% BSA in PBS and keep on ice. Then we performed RNA Library construction with Chromium Single Cell 3′ Reagent Kits v2 following the default process(www.10xgenomics.com). The library was processed on the Illumina HiSeq4000 platform for sequencing with 150 bp paired-end reads.

### Data processing of single-cell RNA-seq from Chromium system
Cell ranger 2.1.1(http://10xgenomics.com) perform quality control and read counting of Ensemble genes with default parameter (v2.1.1) by mapping to hg19 human genome. We exclude poor quality cells after the gene-cell data matrix was generated by cell ranger software using the Seurat package (v3.2.4)[54]. Only cells that expressed more than 800 genes and less than 5000 genes were considered, and only genes expressed in at least 3 single cells were included for further analysis. Cells with the mitochondrial gene percentages over 20% were discarded. In total, 23479 genes across 70036 single cells left for subsequent analysis. The data were normalized to a total of 1e4 molecules per cell for the sequencing depth by using Seurat package. Batch effect was mitigated by using R package batchelor (v1.2.4)[55].

### 10x Genomics Visium sample preparation
Fresh GW12 and GW19 human cerebellum tissue ware embedded in Optimal Cutting Temperature (SAKURA) and frozen in dry ice-ethanol mixture. Frozen tissue block was cut into 10 µm sections in a cryostat (Lecai CM3050 S) and mounted on Tissue Optimization Slides and Gene Expression Slides (10x Gemonics).

Permeabilization time was optimized following manufacturer's instructions (10x Genomics, Visium Spatial Tissue Optimization, CG000238 Rev D) and 25 minutes permeabilization time was chosen for the Visium Spatial Gene Expression workflow. H&E-stained tissue sections were prepared following manufacturer's instructions (10x Genomics, Methanol Fixation, H&E Staining & Imaging for Visium Spatial Protocols, CG000160 Rev B). H&E images and Tissue optimization images were taken using Olympus FV3000 imaging system with Olympus DP80 CCD camera and an Olympus 10X/0.40 objective.

Tissue slices were then processed for gene expression experiment following manufacturer's instructions (10x Genomics, Visium Spatial

Gene Expression Reagent Kits, CG000239 Rev D). In brief, H&E-stained tissue sections were permeabilized for 25 minutes and followed with reverse transcription, second strand synthesis and denaturation. qPCR experiment was processed using KAPA SYBR FAST kit (KAPA Biosystems) and QuantStudio 6 Flex system (ThermoFisher). The cDNA amplification cycle number was determined by ~25% of the peak fluorescence value. The final libraries were processed on Illumina HiSeq Xten system for sequencing of 150 bp pair-end reads.

## Spatial transcriptomics data processing and analysis

Spaceranger (version 1.3.0) software from 10X Genomics was used to perform process, alignment, tissue detection and barcode/UMI counting against the human hg19 reference genome for each spot on the Visum spatial transcriptomic array. The raw UMI count matrix, images, spot-image coordinates and scale factors were imported to R Seurat package, we analyzed the spatial transcriptome data followed the Seurat vignettes of analysis, visualization and integration of 10x Visum datasets. Initially, the regularized negative binomial regression 'SCTransform' function was applied to normalize expression values for total UMI count per cell, this normalization method is better than log-normalization to count for variability in total spot RNA content. Similar to scRNA-seq data analysis, we proceed to run the 'RunPCA', 'FindNeighbors','FindClusters' and 'RunUMAP' functions to do dimensionality reduction and clustering on the RNA expression data.

## Identification of cell types and subtypes by dimensional reduction

Seurat package (v3.2.4) was used to perform linear dimensional reduction. 1500 highly variable genes were selected as input for principal component analysis (PCA). Strongly PC1-24 were used for UMAP to cluster the cells by FindClusters function with resolution 2. Clusters were identified by the expression of known cell-type markers and GO analysis. And markers *MKI67, MFAP4, CBLN1, EOMES, PAX2, PCP4, GDF10, AQO4, OGN, OLIG1, MBP, AIF1, NKG7, MPZ, CLDN5* were used to identify major cell types of cerebellum as Progenitor cells, Granule cells, ECN, UBC, GABAergic neurons, Purkinje cells, Bergmann glial cells, Astrocytes, Meninges, OPC, Oligodendrocytes, Microglia, T cells, Schwann cells, Endothelial cells, respectively.

## Identification of differentially expressed genes among clusters

The differentially expression genes (DEGs) of each cluster were identified by FindAllMarkers function (thresh.use =0.25, test.use = "wilcox") using Seurat R package. Wilcoxon rank sum test (default), and genes with average expression difference > 0.5 natural log with p < 0.05 were selected as marker genes. Enriched gene ontology terms of marker genes were performed using Metascape[56] (http://metascape.org). All the graphs used to show DEGs patterns were plotted by ggplot2 (v3.3.0) or pheatmap (v1.0.12) R package.

## Constructing single cell trajectories in the cerebellum

We used scVelo (v0.2.2) python package[57] to calculate RNA velocity on data. First, we use velocyto run10x to run the counting directly on one or more cellranger output folders (http://velocyto.org/velocyto.py/) by mapping to hg19 human genome. Then we run the RNA velocity calculation following the instructions described on the website (https://scvelo.readthedocs.io/index.html). "scv.pl.velocity_embedding_stream" and "scv.pl.scatter" were applied to show the results of velocyto.

The Monocle 2 R package (version 2.6.4) and Monocle 3 R package(version 0.2.3) were applied to construct single cell pseudo-time trajectory to discover developmental transitions[58–61]. We used highly variable genes identified by Seurat to sort cells in pseudo-time order. The actual gestational time of each cell informs us which state of cells are at beginning of pseudo-time in the first round of "orderCells". We then call "orderCells" again, passing this state as the root_state

argument. "DDRTree" and "UMAP" were applied to reduce dimensional space and the minimum spanning tree on cells was plotted by the visualization functions "plot_complex_cell_trajectory" or "plot_3d_cell_trajectory" for Monocle 2 and Monocle 3, respectively.

## Regulatory Network Comparison in cerebellum

We used pySCENIC(v0.10.3) package[62] for simultaneous gene regulatory network reconstruction and cell-state identification from single-cell RNA-seq data. First, we screened out the tf in our data with hg19_tf list and used GRNBoost2 algorithm to generate of co-expression modules. Then we performed regulon prediction aka cisTarget from CLI and applied "AUCell" to identify cells with active gene sets in single-cell RNA-seq data. After that, we applied scanpy (v1.4.2) python package and pheatmap (v1.0.12) R package to visualize the results of pySCENIC.

## Spatial-specific gene modules enrichment for RL to EGL/EGL to IGL trajectory

To identify key genes changes along RL to EGL/EGL to IGL trajectory, separately. We first defined a pseudospatial-axis along RL to EGL/ELG to IGL by taking spatial information of cells derived from spatial transcriptomic data as reference and then modeled gene expression as a smooth function with pseudospatial-axis by applying a vector generalized additive model (VGAM) with R package VGAM[63,64]. In brief, we fitted gene expression as a smooth function with pre-defined RL to EGL/ EGL to IGL pseudospatial-axis with the vector generalized additive model (VGAM). Heatmaps were then applied for visualization of gene expression branched heatmaps.

## Comparisons of patterns between human and mouse

Seurat (v3.2.4) was a method for integration of large datasets of human and mouse cerebellum RNA expression. We first used "merge" to merge human and mouse Seurat Object without any treating, then use function "SplitObject" to separate the merge data into a list and applied the human data list as the reference.list. Then We calculate the anchors between human and mouse data by "FindIntegrationAnchors" with parameter dims=1:30 and integrated the two list into a new merge data with "IntegrateData". We showed the comparisons of human and mouse with riverplot (v0.6) package.

## Assessing expression patterns of cerebellar disease-associated genes

We first curated gene lists associated with different types of cerebellar diseases, including spinocerebellar ataxia, congenital cerebellar hypoplasia, cerebellar ataxia, pontocerebellar hypoplasia, intellectual disability, Joubert syndrome, autism, Parkinson, Alzheimer and medulloblastoma. The Autism gene list comprised genes from the syndromic category in the SFARI Gene (https://gene-archive.sfari.org/database/gene-scoring/), a comprehensive database that contains gene associated with autism risk. The Alzheimer and Parkinson gene lists were collected from https://www.labome.com/method/Alzheimer-s-Disease-Genes.html and https://www.healthline.com/health/parkinsons/is-parkinsons-hereditary, respectively. The medulloblastoma gene list was compiled from three published studies[65–67]. The other gene lists were obtained from the ClinVar database (https://ftp.ncbi.nlm.nih.gov/pub/clinvar/vcf_GRCh38/clinvar_20201219.vcf.gz) by extracting genes with known pathogenic mutations for each disease. The full disease gene lists were provided in Supplementary Data 14.

We next assessed the expression patterns of those disease-associated genes in each cell type. Only genes expressed within at least 20% of cells of at least one cell subtypes were retained for the downstream analysis. We first summarized the expression of all disease genes across cell types by computing two matrices, one represented the fraction of cells in each cell type that had non-zero expression of

each gene and the other one represented the average expression value of each gene in each cell type. We then computed a matrix of gene expression scores for all genes across each cell type as the product of these two matrices. To quantify the aggregated expression level of gene lists in each cell type, we further calculated the average expression scores for each gene list across each cell type and visualized their patterns in a heatmap. The selected disease-associated genes were also visualized by using the gene expression scores in each cell type described above, with the adjusted P value calculated from Wilcoxon test and Benjamini-Hochberg multiple hypothesis correction.

We applied LD score regression (LDSC) to cell type-specific ATAC-seq peaks to identify GWAS traits associated cell types. LDSC used GWAS summary statistics as input and quantified the enrichment of heritability in an annotated set of SNPs. We downloaded the cerebellum-related GWAS summary statistics from https://alkesgroup.broadinstitute.org/sumstats_formatted/ (Alzheimer, Schizophrenia, Bipolar disorder and Autism) and https://zenodo.org/record/3817811 (Parkinson). Additionally, we used two GWAS traits without obvious relationship with brain tissues (Celiac and Lupus, downloaded from https://alkesgroup.broadinstitute.org/sumstats_formatted/) as controls. Followed the LD score regression tutorial (https://github.com/bulik/ldsc/wiki), we first used the HapMap SNPs, precomputed files corresponding to 1000 genomes phase 3 to generate an LDSC model for each chromosome and peak set. Then, the script ldsc.py was used with default parameters, the baseline model and the full GWAS summary statistics above to calculate enrichments. P-values were calculated from z-scores assigned to coefficients and the multiple hypothesis test was corrected using the Benjamini-Hochberg method. Only tests with an adjusted P value less than or equal 0.05 were considered as significant.

## ATAC library preparation for high-throughput sequencing
We aimed to collect only nuclei from Flash Frozen Tissue for ATAC library preparation. We transfer the frozen tissue to a 1.5 ml microcentrifuge tube with chilled forceps, add 500 µl chilled 0.1X Lysis Buffer, homogenize and incubate for 15 min on ice. Add another 500 µl chilled 0.1X Lysis Buffer and mix. The pass the suspension through a 40 µm Flowmi Cell Strainer into a 2-ml tube. Centrifuge at 500 g for 5 min at 4 °C, the resuspended the nuclei pellet with chilled Diluted Nuclei Buffer. Then we performed ATAC Library construction with Chromium Single Cell ATAC Reagent Kits v1.0 following the default process (www.10xgenomics.com). The library was processed on the Illumina Nova-Seq platform for sequencing with 50 bp paired-end reads.

## Quality control of single-cell ATAC-seq
Reads filtering and mapping (mapping to human reference genome hg19) and transposase cut sites identification were performed with the Cell Ranger ATAC Software (v1.2.0). The generated fragment files were passed to the R package SnapATAC (v 1.0.0) and snap file with bin size of 2.5 kb was generated. The produced cell-by-bin count matrix was first converted into a binary matrix by replacing the non-zero items to 1. Then, cells that were outliers for any of the following criterion were considered to be of low quality and discarded: 1) log10-transformed number of unique fragments between 2 and 5 ($\log_{10}$(UMI)); 2) fragments in promoter ratio between 0.15 and 0.6; 3) the number of fragments in peaks between 1000 and 40000; 4) the fraction of fragments in peaks greater than 30%; 5) transcriptional start site (TSS) enrichment score greater than 1.8. Bins overlapping with the ENCODE blacklist were excluded. The filtered binarized cell-by-bin matrix was the loaded to R package Signac (v0.2.5) for downstream analysis.

## Dimension reduction and cell type identification
The R package Signac (v0.2.5)[68] was used for to perform dimension reduction and cell type identification. The filtered cell-by-bin count matrix was first TF-IDF (frequency-inverse document frequency) normalized with function RunTFIDF, followed by SVD (singular value decomposition) linear dimension reduction analysis by function RunSVD based on the top 25% variable peaks identified with function FindTopFeatures by setting min.cutoff to q75. The reduced dimensions were then imported into function RunUMAP for subsequent nonlinear dimension reduction. Clustering was done with function FindNeighbors and FindClusters. Next, gene activity matrix was created by quantifying the activity of each gene in the genome by accessing gene associated chromatin accessibility using function FeatureMatrix. Clusters were then annotated based on the gene activity profile of know cell-type markers.

## Peak calling
For each cluster, peak calling was performed with fragments from cells in the cluster by command MACS2 with parameters '−nomodel −shift 100 −extsize 200 −qval 5e-2 −B −f BED −nolambda −keep-dup all −callsummits' callpeak with MACS2 software (v 2.2.7.1). Peaks from each cluster were then merged and a binarized cell-by-peak matrix was constructed by convert non-zero counts to 1.

## Calculation of differentially accessible peaks
Differentially accessible peaks between clusters were computed with Signac function FindAllMarkers by setting test.us = 'LR' and latent.-vars='peak_region_fragments'. Only peaks with adjusted p-value smaller than 0.05 were considered as differentially accessible peaks.

## Integration of scRNA-seq, spatial transcriptomic and snATAC-seq data
To compare the transcriptomic profiles of 10x Genomics Visium, TF-seqFISH and scRNA-seq dataset, we assembled TF-seqFISH with scRNA and 10x Genomics Visium dataset, separately using the AlignSubspace function of Seurat[54] with shared variable genes in each merged data. Cell clustering and dimensionality reduction were performed with FindClusters and RunUMAP function, respectively. To evaluate cell occupancy in each cluster, we computed the cell ratio of TF-seqFISH to scRNA and 10x Genomics Visium dataset in each cluster after normalizing the total cell numbers, separately. Then we used the same method to cope with scRNA and snATAC dataset. River plots were constructed to illustrate the mapping pattern of cells from each dataset.

## Probe library design for TF-seqFISH
Gene targeted regions for 1085 human TF genes were selected from exons based on hg38 genome release-97. Each gene had 17-32 target regions which were 28-nt in length and had a Tm range from 50 to 80. We run a local BLAST query for each selected sequence against human transcriptome to remove 17-nt off-target sequences. Readout sequences targeted regions were randomly generated sequences that had a GC fraction range from 40% to 60%. A local BLAST against human and mouse genome was performed to select specificity sequences from this randomly generated pool. Any sequences which had a 10-nt match with each other were dropped.

## Probe construction for TF-seqFISH
Gene targeted probes were constructed from complex oligonucleotide pools (Twist Bioscience). We amplified the custom designed oligonucleotide pools by limited-cycle PCR (M0536L, NEB; 31000-T, Biotium). The amplified dsDNA was performed as templates for in vitro transcription started by T7 RNA polymerase (M0251L, NEB). Reverse transcription was followed to generate ssDNA probes (EP0753, Thermo Scientific). The reverse transcription products mix was treated with uracil-specific excision reagent (USER) enzyme (M5505L, NEB) to cleave off 5′forward primer. Alkaline hydrolyzing was processed to remove RNA template with 0.5 N NaOH (S2770, Sigma) and 0.25 M EDTA (15575020, Thermo Scientific) at 95 °C for 20 minutes. Ethanol

precipitation was followed to purify ssDNA probes from the reaction mix. Constructed probes were stored at -80 °C before use. Readout probes were ordered from Integrated DNA Technologies as 5′ modified with Alexa Fluor 647, Alexa Fluor 594 or Alexa Fluor 488.

## Tissue slices preparation for TF-seqFISH

Fresh tissues were fixed in 4% PFA and then sank in 30% sucrose (V900116, Sigma), followed by embedding in O.C.T. Compound (4583, Sakura). The frozen tissue block was cut into 10 μM sections and permeabilized in 70% ethanol at 4 °C and then 10% Sodium dodecyl sulfate (L3771, Sigma) at room temperature. The target probe hybridization was proceeded in a 37 °C oven overnight with first hybridization buffer: 500 ng/μL gene targeted probe; 10% dextran sulfate sodium salt (D8906, Sigma), 40% formamide (47671, Sigma); 2X SSC (93017, Sigma); 2 μM T15 LNA (Takara). After hybridization, the tissue slices were washed in 20% formamide at 37 °C and then treated with 0.1 mg/mL label-X at 37 °C. Then tissue slices were embedding in acrylamide/bis-acrylamide hydrogel (A2917, Sigma) and digested with 10 U/mL protease K (P8107S, NEB). Hydrogel slices were stored in 2X SSC with 0.4 U/μL RNase inhibitor (AM2696, Thermo Scientific) at 4 °C before imaging.

## TF-seqFISH imaging

Tissue sections embedded in hydrogel were attached to a flow-cell chamber designed in an appropriate size. The automated fluidics system was connected with a spinning disk imaging system (Dragonfly 500, Andor technology ltd). After ROIs (region of interests) were selected, 22 hybridization-imaging cycles were performed. During each hybridization-imaging cycle, hybridization buffer containing 50 nM readout probe, 8% ethylene carbonate (E26258, Sigma), 10% dextran sulfate sodium salt (D4911, Sigma) and 0.4 U/μL RNase inhibitor in 2X SSC was pumped into the flow-cell chamber, followed with 10 μg/μL DAPI solution (62248, Thermo Scientific), 20% formamide washing solution and anti-bleaching solution containing 8% (w/v) D-(+)-Glucose (G8270, Sigma), 1 mg/mL glucose oxidase (G2133, Sigma), 10 μg/mL catalase (C3155, Sigma), 0.4 U/μL RNase inhibitor. Images of 488, 637 and 594 channels for selected FOVs were taken in each hybridization-imaging cycle as signal images. Images of 405 channel were taken for cell-segmentation and horizontal-shift correction.

## Image analysis preprocessing

**Image registration.** Each round of images consisted of three signal channels and a 405-nm channel for cell segmentation. For each FOV, all serial hybridization rounds were registered to the initial round using DAPI image. Common feature points of the two images were aligned to get the final transformation matrix.

**Image processing.** Uneven illumination correct and background removal were performed by subtracting the background image. Then two algorithms were applied to remove as much noise as possible. Non-Local Means estimates the pixel value using the weighted average value of which has similar neighborhood structure while Total Variation assumed that signals with excessive and possibly spurious detail have high total variation to remove unwanted noise. After these, most technical errors were corrected and the next step was to reconstruct signals. Each image was deconvolved with an estimated 7*7-pixels point spread function (PSF) from beads image.

**Cell segmentation.** All cells were segmented by a pre-trained Res-UNet model using the DAPI images and Nissl images. The output of this model was the probability of cell instance for post-process algorithm. ROI results were refined in ImageJ manually.

## Barcode calling

Another pre-trained Res-UNet model was implemented to detect every single spot. Once all potential spots in all rounds were obtained, they were reorganized by pseudo-colors and barcoding round. Nearest neighbors within 1.2-pixel radius were found out for every spot in other three barcode channels. While the combination of mutual nearest neighbor spots match to a unique barcode, these spots were added to the barcode set. For the other spots which were matched to multiple barcodes, we kept the point sets with the minimum distances, followed by matching the spots to the nearest target barcode whose hamming distance is no more than one. The rest of the ambiguous points were dropped. We repeated this procedure using each of four barcode channels as a seed, and only retained the gene called with more than three seeds. For the cells in the overlapped area shared by adjacent FOVs, one cell with less detect mRNAs would be removed if this cell was observed twice. And the expression of each cell could also be weighted by its neighbors to enhance the data quality which can eliminate the dropout effects or other unwanted noise for downstream analysis.

## Immunofluorescent staining

Tissue samples were fixed overnight in 4% paraformaldehyde, cryo-protected in 30% sucrose, and embedded in optimal cutting temperature (Thermo Scientific). Thin 40 μm cryosections were collected on superfrost slides (VWR) using Leica CM3050S cryostat. For immunohistochemistry, heat-induced antigen retrieval was performed in 10 mM sodium citrate buffer, pH 6. Primary antibodies: rabbit anti-ASCL1(MASH1) (1:200, Abcam ab213151), rabbit anti-PTPRZ1(1:200,Sigma HPA015103), rabbit anti-TTYH1 (1:300, Sigma-Aldrich HPA023617), mouse anti-CALB1(CALBINDIN)(1:200, Abcam ab9481-500), rabbit anti-RORB(1:250, Invitrogen PA5-28742), rabbit anti-PAX6(1:500, BioLegend 901301), and were diluted in blocking buffer containing 10% Donkey Serum, 0.5 % Triton-X100 and 0.2% gelatin diluted in PBS at pH=7.4. Binding was revealed using an appropriate Alexa Fluor™ 488, Alexa Fluor™ 594, and Alexa Fluor™ 647 fluorophore-conjugated secondary antibody (Life Technologies). Cell nuclei were counter-stained using DAPI (Life Technologies). Images were collected using an Olympus FV1000 Confocal microscope.

## Plasmids and in-utero electroporation

*ARHGAP11B* genes were cloned into pEGFP-C1 vector. Electroporation was performed as previously described[69]. Briefly, timed pregnant CD-1 mice (E11.5) were deeply anesthetized with isoflurane, and the uterine horns were exposed through a midline incision. 1 μl of plasmid DNA (2-3 μg/μl) mixed with Fast Green (Sigma) was manually microinjected into the brain lateral ventricle through the uterus, using a bevelled and calibrated glass micropipette (Drummond Scientific) followed by five 50-ms pulses of 35 mV with a 1 s interval delivered across the uterus with two 9-mm electrode paddles positioned on either side of the head (BTX, ECM830).

## Reporting summary

Further information on research design is available in the Nature Portfolio Reporting Summary linked to this article.

# Data availability

The scRNA-seq data and ATAC-seq data generated in this study have been deposited in the Gene Expression Omnibus (GEO) database under accession code GSE165657 and in the Genome Sequence Archive (GSA) under accession number HRA000780. The disease risk genes used in this study are available in the SFARI database [https://gene-archive.sfari.org/database/gene-scoring/] and the ClinVar database [https://ftp.ncbi.nlm.nih.gov/pub/clinvar/vcf_GRCh38/clinvar_20201219.vcf.gz]. Source data are provided with this paper.

# Code availability

All analyses in our study were performed with open-source packages as described in the Methods section.

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

## Acknowledgements

This work was supported by the National Natural Science Foundation of China (NSFC) (32122037, 32222032, 32192411, 32000570), National Basic Research Program of China (2019YFA0110101), Science and Technology Innovation 2030—Brain Science and Brain-inspired Intelligence Project of China (2022ZD0207700, 2022ZD0211900), Collaborative Research Fund of Chinese Institute for Brain Research, Beijing (2020-NKX-PT-03), CAS Project for Young Scientists in Basic Research (YSBR-013), the New Cornerstone Science Foundation, the Fundamental Research Funds for the Central Universities and Changping Laboratory.

## Author contributions

Q.W., X.W. and S.Z. conceived the project and designed the experiments. S.Z. and J.Z. performed the single-cell RNA-seq experiment. S.Z. and Y.L. performed the scATAC-seq experiment. Y.G., L.H., H.D. and S. Z. performed spatial transcriptome experiments and data analysis. S.Z. and X.J. analyzed the RNA-seq data. M.W. and Y.S. analyzed the ATAC-seq data. X.Z., Y.C. and T.L. prepared the samples. Y.G. and B.W. performed the tissue section, immune-staining. S.Z., Y.G. and J.Z carried out animal experiments. S. Z. and Q. W. wrote the manuscript and all authors edited and proofed the manuscript.

## Competing interests

The authors declare no competing interests.
