## [Peer Review File · Nature Communications]

Single-cell epigenomics and spatiotemporal transcriptomics reveal human cerebellar developmentREVIEWER COMMENTS

Reviewer #2 (Remarks to the Author):

Zhong et al. analyzed the epigenomic and transcriptomic profile of the human developing cerebellum from GW12-27. The cellular atlas revealed cell type diversity, spatial organization, and inferred potential regulatory mechanisms of neuronal and glial cell differentiation. A new marker, RORB, was identified for Purkinje cell subgroups and the gene ARHGAP11B was found to play a role in the evolutionary expansion of the human cerebellum. This study will be a significant resource for the field. However, I would suggest clarifying a few points as detailed below:

Major comments:

Figure 1C: Could you clarify how different regions were defined? Was it based on marker gene expression or a human brain anatomy atlas?

Figure 1F: Integration of snRNA-seq, spatial transcriptomic and snATAC-seq data exhibits correspondences between clusters identified in each dataset. Could you provide detailed description of how the integration analysis was conducted? It will help to understand how cell clusters identified in 10x Genomics Visium, TF-seqFISH, and scRNA-seq correspond to each other.

Figure 2D: You report that FOXJ1 is enriched in progenitor 6 and 8, yet FOXJ1 is not depicted in the heatmap. Could this be rectified?

Figure 3C: You state "RORA is a classical marker of Purkinje cells, but RORB is not. RORB+ cells were only a subset of RORA+ cells (Figure 3C and S9D)." It would be more informative to compute the proportion of cells expressing each. Is there any difference in the timing of expression of both genes? Are RORA and RORB different forms of ROR family and how similar are these two genes in sequence?

Figure 4: Regarding granule cell sub-clustering, how does cell distribution change over the developing time? Projecting this information onto the UMAP in Figure S10A could provide more insights.

Figure 4I and L: You identify 20 gene modules that show spatial cascade profiles, but it's unclear how the x-axis was defined in the heatmap. Could you provide a detailed description in the methods section?

Figure 6B: Are the genes shown here as a subset of those in Figure S12A? Did you conduct enrichment analysis separately on snRNA-seq and snATAC-seq? Have you observed SNPs respectively located in coding and non-coding regions link to same genes? If yes, are those SNPs associate with the same disease phenotype?

Minor comments:

Figure S12A: The gene labels on the heatmap related to ASD appear to have shifted.

In Figure S9D, no scale bar providing gene expression level information was shown.

Reviewer #3 (Remarks to the Author):

The manuscript by Zhong et al. offers detailed analyses of human cerebellum development, integrating single-cell transcriptomics, chromatin accessibility, and spatial transcriptomics to enrich

our understanding of cerebellar development. Particularly noteworthy is the study's illumination of the hierarchical differentiation processes of progenitor cells into various cell types. The analysis focused on how RL cells differentiating into EGL cells and EGL cells differentiating into IGL cells are impressive. Although the manuscript is well-conceived and -written, I would offer the following comments and suggestions:

Major comments/questions:

1. Page 6, Lines 166-170: The authors claim that 'genes belonging to the same GO term in some cases are involved in different modules.' This is intriguing but somewhat confusing. Are there common genes in different modules related to neuron migration? Or all the genes regulating neuron migration in each region are different?
2. Page 7, Lines 195-197: The manuscript states that ' Moreover, genes related to glial differentiation, such as TTYH1, GFAP and FOXJ1, were expressed at high levels in Progenitor 6 and Progenitor 8, suggesting the glial cell fate of these progenitors (Figure 2D, E).' The glial cell fate from VZ lineage isn't shown in the Figure 2E. Can you also elaborate on the timeline of VZ progenitors differentiate into each glial subtype in the human developing cerebellum?
3. Pages 6-7: Does the RL lineage differentiation follow the sequence of eCN, granule cell, and UBCs in order? Is this timeline different between humans and mice?
4. Pages 9-10: The discovery of different Purkinje subtypes is potentially significant. However, given the limited information, understanding the developmental differences of these subtypes is challenging. Do Purkinje cells exhibit varying maturation statuses in different subtypes within the developing cerebellum?
5. Pages 10-11: The development and migration of granule cells are perhaps the most exciting aspects of this study, especially the regulation of EGL migration to form IGL. The interactions of these migrating granule cells with the Purkinje Layer and Bergmann glial cells are still unclear. Do the multi-omic datasets reveal any new insights into these interactions or regulations?
6. Pages 13-14: The analysis of cell types and disease-associated SNPs in coding and non-coding regions is impressive. Are there any differences between these two datasets? Do some genes both mutated in coding and non-coding regions?

Minor points:

1. Page 2, Line 45-48: 'we combined single-cell transcriptomics, spatial transcriptomics and chromatin accessibility states...', the authors used the description of 'spatial transcriptomics and chromatin accessibility states' misled the concept that you use spatial ATAC-seq data, please correct the description.
2. Page 2, Line 48-49: 'Our multiomic data revealed that combinations of transcription factors at CREs play roles in ...'. Please correct the word 'at'.
3. In Figure 1D, the gene name of granule cell group was missing.
4. In Figure 2G, some gene names are unclear.
5. Page 7, Line 196: FOXJ1 seems not in Figure 2D,E.

We thank the reviewers for their thoughtful comments and suggestions, which overall are very positive and constructive. We believe we have addressed their questions and concerns, as described below. In this document, original comments made by the reviewers are in *blue*, and our responses are in black.

REVIEWER COMMENTS

Reviewer #2 (Remarks to the Author):

Zhong et al. analyzed the epigenomic and transcriptomic profile of the human developing cerebellum from GW12-27. The cellular atlas revealed cell type diversity, spatial organization, and inferred potential regulatory mechanisms of neuronal and glial cell differentiation. A new marker, RORB, was identified for Purkinje cell subgroups and the gene ARHGAP11B was found to play a role in the evolutionary expansion of the human cerebellum. This study will be a significant resource for the field. However, I would suggest clarifying a few points as detailed below:

Response: We thank the reviewer for the insightful and supportive comments. We have carefully addressed all of the concerns raised by the reviewer (see below).

Major comments:

Figure 1C: Could you clarify how different regions were defined? Was it based on marker gene expression or a human brain anatomy atlas?

Response: We thank the reviewer for the very useful comment. We defined the exact region information based on the human cerebellum anatomy structures in text books and references (Bayer and Altman,2002) . We clustered the spatial results and found the marker genes of each cluster. We then plot the marker gene of spatial transcriptomic results to visualize their area specificity, and also search the possible function and cell type association of these marker genes to further validate the regional identity. For example, in GW12, EGL (external granule cell layer) is reviewed in anatomy book and granule cell marker *MGP* was also highly and specifically expressed in this area. Thus, both anatomy structure and marker gene expression show the same result (Response Figure 1).

Action taken:

- 1) We double checked the defined region of Figure 1C, D to make sure we defined the exact region information.
- 2) We revised the manuscript with the description of how different regions were defined to make the text more clearly (Page 5, Lines 143-146) as follow.

“We defined the exact region information based on the human cerebellum anatomy structures (Bayer and Altman,2002) and validated the marker gene expression in the spatial transcriptomic results.”

Response Figure 1

Figure 1F: Integration of scRNA-seq, spatial transcriptomic and snATAC-seq data exhibits correspondences between clusters identified in each dataset. Could you provide detailed description of how the integration analysis was conducted? It will help to understand how cell clusters identified in 10x Genomics Visium, TF-seqFISH, and scRNA-seq correspond to each other.

Response: We are grateful for the reviewer’s constructive comments and we apologize for not explaining this clearly. We now added the detailed description of how the integration analysis was conducted in the revised methods, we hope that would be helpful to understand the clusters identified in 10x Visium, TF-seqFISH, scRNA-seq and snATAC correspond to each other.

Action taken:

We revised the method with intergraiton of scRNA-seq, spatial transcriptomic and snATAC-seq data in the manuscript (Page 29, Lines 796-805) as follow.

“Integration of scRNA-seq, spatial transcriptomic and snATAC-seq data

To compare the transcriptomic profiles of 10x Genomics Visium, TF-seqFISH and scRNA-seq dataset, we assembled TF-seqFISH with scRNA and 10x Genomics Visium dataset, separately using the AlignSubspace function of Seurat (Stuart et al.,2019) with shared variable genes in each merged data. Cell clustering and dimensionality reduction were performed with FindClusters and RunUMAP function, respectively. To evaluate cell occupancy in each cluster, we computed the cell ratio of TF-seqFISH to scRNA and 10x Genomics Visium dataset in each cluster after normalizing the total cell

numbers, separately. Then we used the same method to cope with scRNA and snATAC dataset. River plots were constructed to illustrate the mapping pattern of cells from each dataset.”

Figure 2D: You report that FOXJ1 is enriched in progenitor 6 and 8, yet FOXJ1 is not depicted in the heatmap. Could this be rectified?

Response: We thank the reviewer for pointing this out and apologize for not clearly depicting the FOXJ1 pattern in the heatmap in Figure 2D. We further checked our results and found FOXJ1 was highly expressed in Progenitor 8 (Response Figure 2, Figure S6D, and Table S7 Line 776). Therefore, we assigned FOXJ1 is a marker gene for Progenitor 8 and we updated Figure 2D (showing as Response Figure 3).

In the manuscript, we claimed that “TTYH1, GFAP and FOXJ1, were expressed at high levels in Progenitor 6 and Progenitor 8” (Page 7 Line 195-197) which means TTYH1 and GFAP are the marker genes for Progenitor 6, the gene FOXJ1 is the marker gene for Progenitor 8. FOXJ1 is a member of the Forkhead/winged-helix (Fox) family of transcription factors, which is required for the differentiation of the cells acting as neural stem cells which participate in gliogenesis and give rise to astrocytes and oligodendrocytes(Jacquet et al.,2009). So in the revised manuscript, we make this clear.

Response Figure 2

Response Figure 3 / new Figure 2D

Action taken:

- 1) We checked the expression of FOXJ1 for VZ progenitor in Figure S6D, Table S7 and the Response Figure 2,3 and make sure the FOXJ1 was the marker gene for Progenitor 8.
- 2) We labeled the *FOXJ1* in new Figure 2D and revised the text in the manuscript to make it more readable (Page 7, Lines 199-200).

Figure 3C: You state “RORA is a classical marker of Purkinje cells, but RORB is not. RORB+ cells were only a subset of RORA+ cells (Figure 3C and S9D).” It would be more informative to compute the proportion of cells expressing each. Is there any difference in the timing of expression of both genes? Are RORA and RORB different forms of ROR family and how similar are these two genes in sequence?

Response: We thank the reviewer for the constructive comments on improving our study. Following the reviewer’s suggestion, we checked the proportion of RORA⁺ and RORB⁺ Purkinje cells in the Purkinje cell group. We found that the proportion of Purkinje cells expressing RORA and RORB is 94.07% and 67.68% in the single-cell data. In the RORB⁺ Purkinje cell group, 98.56% were RORA⁺, which indicated that RORB⁺ Purkinje cells were a subset of RORA⁺ Purkinje cells.

Then we checked the proportion of RORA⁺ and RORB⁺ Purkinje cells in different developmental time. We found both RORA⁺ and RORB⁺ groups maintained a stable proportion during the development of cerebellum in human brain as the Response Figure 4 shown.

RORA and *RORB* genes coding retinoic acid receptor-related orphan receptor (ROR) α and β belong to the nuclear receptors superfamily which is composed of 48 members in humans(Zhang et al.,2015). Following the reviewer’s suggestion, we compared the protein sequence of RORA and RORB with ClustalW online software (<http://www.clustal.org/clustal2/>). A multiple sequence alignment was performed as Response Figure 5 shown. The aligned score between RORA (580aa) and RORB (470aa) protein sequence is 58.9362. In Zhang’s work, they compared and conclude that RORA shared high sequence identity and conserved domains with RORB(Zhang et al.,2015). We added this information in the section of discussion, regarding to the RORA and RORB expression in the Purkinje cells (Pages 16-17, Lines 455-456).

Response Figure 4

CLUSTAL 2.1 multiple sequence alignment

```

RORA      MNEGAPGSDLEEARVPSIMGHCLRTGQARMSATPTPAGEGARSSSTCSSLSRLFWSQ
RORB      -----

RORA      LEHINWDGATAKNFINLREFFSFLPALRKAQIEIIPCKICGDKSSGIHYGVITCEGCKG
RORB      -----MCENQLTKADATAQIEVIPCKICGDKSSGIHYGVITCEGCKG
          .: * ,*****:*****:*****:*****:*****:*****:*****

RORA      FFRRSQSNATYSCPRQKNCLIDRTSRNRCQHCLQKCLAVGMSRDAVKFGRMSKKQRDS
RORB      FFRRSQNNASYSCPRQRNCLIDRTNRNRCQHCLQKCLALGMSRDAVKFGRMSKKQRDS
          *****.*:*****:*****.*:*****:*****:*****:*****:*****

RORA      LYAEVQKHRMQQQRDHQQPGEAEPLTPTYNISANGLTELHDDLSDNYIDGHTPEGSKAD
RORB      LYAEVQKHQRLQEQ-RQQSGEAEALARYSSISNGLSNLNNETSGTYANGHVIDLPK
          *****: : *:: :***,***.*: .*. * . . . : : : . . : . .

RORA      SAVSSFYLDIQSPDQSGLDIN---GIKPEPICDYTPASGFFPYCSFTNGETSPTVSMAE
RORB      SEGYYNVDSGQSPDQSGLDMTGIKQIKQEPYDLTSVPNLFTYSSFNNGQLAPGITMTE
          * . *****: . ** ** * * . . . : * . * . * : * : * : *

RORA      LEHLAQNIKSHLETQYLREELQQITWQTFLQEEIENYQNKQREVMWQLCAIKITEAIQ
RORB      IDRIAQNIKSHLETQYTMEEHLQLAWQHTYEEIKAYQSKSREALWQCAIQITHAIQ
          : : : * * * * * * * * * * * * * * * * * * * * * * * * * * * * * * *

RORA      YVVEFAKRIDGMELCQNDQIVLLKAGSLEVVFIRMCRAFDSQNTVYFDGKYASPDVFK
RORB      YVVEFAKRITGMELCQNDQILLKSGCLEVVLVRCRAFNLNNTVLFEGKYGGQMFK
          ***** *****:*****:*****:*****: . ***** * : * * . : : *

RORA      SLGCEDFISVFVFGKSLCSMHLTEDEIALFSAFVLMADRSWLQEKVKIEKLQKIQILA
RORB      ALGSDDLVNEAFDFAKNLCSLQLTEEEIALFSSAVLISPRAWLIEPRKVQLQEKIYFA
          : * . : * : . . * : * . * . * : * : * : * : * : * : * : * : * : * : *

RORA      LQHLQKNHREDGILTKLICKVSTLRALCGRHTEKLMAFKAIYPDIVRLHFPPLYKELFT
RORB      LQHVIQKHLDDDELAKLIAKIPITITAVCNLHGEKLVQKQSHPEIVNTLFPPLYKELFN
          *****: * * : * * . * : * : * . * * * * * * : * * . * * * * *

RORA      SEFEPAMQIDG
RORB      PDCATGCK---
          .: . . :
    
```

Response Figure 5

Action taken:

- 1) Following the Reviewer’s suggestion, we checked the proportion of RORA⁺ and RORB⁺ groups in the Purkinje cells to show that RORA is a classic marker for most

Purkinje cells and RORB⁺ Purkinje group is a subset of RORA⁺ Purkinje cells along the development in human embryonic cerebellum.

- 2) We compared the sequences of RORA and RORB and added the discussion in the manuscript (Pages 16-17, Lines 455-456).

Figure 4: Regarding granule cell sub-clustering, how does cell distribution change over the developing time? Projecting this information onto the UMAP in Figure S10A could provide more insights.

Response: We are grateful for the reviewer's question. Following the reviewer's suggestion, we checked the cell distribution change over the developing time and projected this information onto the UMAP in Response Figure 6/ new Figure S10A as follow. We found that subtypes of granule cells showed different maturation among all the week stages in the developing cerebellum, indicating that the differences of subtypes were not come from real developing time (gestational weeks).

Response Figure 6/ new Figure S10A

Action taken:

- 1) We added a UMAP plot of real developmental time and pseudotime in granule cell subtypes as new Figure S10A to show the cell distribution change of different subtypes over the developing time.
- 2) We revised the text and figure legends in the manuscript accordingly (Page 17, Line 474-477; Page 42, Lines 1261-1262).

Figure 4I and L: You identify 20 gene modules that show spatial cascade profiles, but it's unclear how the x-axis was defined in the heatmap. Could you provide a detailed description in the methods section?

Response: We apologized for not defining the x-axis in the heatmap clearly, and we revised in the methods accordingly. We added how we defined the x-axis in the heatmap in Figure 4 in the revised methods, we hope that would be helpful for readers to understand the way how spatial-specific gene modules promote RL progenitors and EGL cells differentiate into granule cells.

Action taken:

We revised the method with spatial-specific gene modules enrichment for RL to EGL/EGL to IGL trajectory in the manuscript (Pages 24-25, Lines 679-687) as follow.

“Spatial-specific gene modules enrichment for RL to EGL/EGL to IGL trajectory

To identify key genes changes along RL to EGL/EGL to IGL trajectory, separately. We first defined a pseudospacial-axis along RL to EGL/ELG to IGL by taking spatial information of cells derived from spatial transcriptomic data as reference and then modeled gene expression as a smooth function with pseudospacial-axis by applying a vector generalized additive model (VGAM) with R package VGAM(Yee,2015,Yee and Wild,1996). In brief, we fitted gene expression as a smooth function with pre-defined RL to EGL/ EGL to IGL pseudospacial-axis with the vector generalized additive model (VGAM). Heatmaps were then applied for visualization of gene expression branched heatmaps.”

Figure 6B: Are the genes shown here as a subset of those in Figure S12A? Did you conduct enrichment analysis separately on snRNA-seq and snATAC-seq? Have you observed SNPs respectively located in coding and non-coding regions link to same genes? If yes, are those SNPs associate with the same disease phenotype?

Response: We thank the reviewer for the insightful comments. First, the genes in the Figure 6B are not a subset of Figure S12A. We conducted enrichment analysis for scRNA-seq data for the coding region shown in the Figure 6A (left part) and Figure S12A and snATAC-seq data for the non-coding region shown in the Figure 6A (right part) and Figure 6B. Sorry for not explaining it clearly. We revised the descriptions in the text to make it more readable (Page 14, Lines 388-395).

We are grateful for the reviewer’s suggestions and completely agree that the analysis of whether the SNPs respectively located in coding and non-coding regions link to same genes and the same disease phenotype would strengthen the depth of our findings. Following the reviewer’s suggestion, we calculated the SNPs both located in coding and non-coding regions link to the same gene. We found that 264 genes with coding and non-coding SNPs at the same time in our datasets, 46.59% of them associated with different disease phenotype. We have added a new table as Table S15 for this information.

Action taken:

- 1) We revised the descriptions of the enrichment analysis conducted separately on snRNA-seq and snATAC-seq in the text (Page 14, Lines 388-391).
- 2) Following the reviewer's suggestion, we analyzed the genes with SNPs in both coding and non-coding regions at the same time. We observed 1619 genes in Figure 6A, and found that 1355 genes (83.7%) with SNPs either in the coding regions or non-coding regions, while 264 genes (16.3%) with SNPs both in coding and non-coding regions. Among these 264 genes, 46.59% of them associated with different disease phenotype. We added information of these 264 genes in the new Table S15 and revised the manuscript accordingly (Page 14, Lines 391-395).

Minor comments:

Figure S12A: The gene labels on the heatmap related to ASD appear to have shifted.

Response: Following the reviewer's suggestion, we revised the gene labels on the heatmap related to ASD in Figure S12A.

In Figure S9D, no scale bar providing gene expression level information was shown.

Response: Following the reviewer's suggestion, we added scale bar for gene expression in Figure S9D.

References.

1. Bayer, S.A., and Altman, J. (2002). Atlas of human central nervous system development (CRC Press).
2. Stuart, T., Butler, A., Hoffman, P., Hafemeister, C., Papalexi, E., Mauck, W.M., Hao, Y.H., Stoeckius, M., Smibert, P., and Satija, R. (2019). Comprehensive Integration of Single-Cell Data. *Cell* 177, 1888-+. 10.1016/j.cell.2019.05.031.
3. Jacquet, B.V., Salinas-Mondragon, R., Liang, H., Therit, B., Buie, J.D., Dykstra, M., Campbell, K., Ostrowski, L.E., Brody, S.L., and Ghashghaei, H.T. (2009). FoxJ1-dependent gene expression is required for differentiation of radial glia into ependymal cells and a subset of astrocytes in the postnatal brain. *Development* 136, 4021-4031. 10.1242/dev.041129.
4. Zhang, Y., Luo, X.Y., Wu, D.H., and Xu, Y. (2015). ROR nuclear receptors: structures, related diseases, and drug discovery. *Acta Pharmacol Sin* 36, 71-87. 10.1038/aps.2014.120.
5. Yee, T.W. (2015). Vector generalized linear and additive models : with an implementation in R (Springer).
6. Yee, T.W., and Wild, C.J. (1996). Vector generalized additive models. *J Roy Stat Soc B* 58, 481-493.

Reviewer #3 (Remarks to the Author):

The manuscript by Zhong et al. offers detailed analyses of human cerebellum development, integrating single-cell transcriptomics, chromatin accessibility, and spatial transcriptomics to enrich our understanding of cerebellar development. Particularly noteworthy is the study's illumination of the hierarchical differentiation processes of progenitor cells into various cell types. The analysis focused on how RL cells differentiating into EGL cells and EGL cells differentiating into IGL cells are impressive. Although the manuscript is well-conceived and -written, I would offer the following comments and suggestions:

Response: We thank the reviewer for the insightful and supportive comments. We have carefully addressed all of the concerns raised by the reviewer (see below).

Major comments/questions:

1. Page 6, Lines 166-170: The authors claim that 'genes belonging to the same GO term in some cases are involved in different modules.' This is intriguing but somewhat confusing. Are there common genes in different modules related to neuron migration? Or all the genes regulating neuron migration in each region are different?

Response: We apologized for not describing the Figure 1I clearly. In this panel, we used the WGCNA package to enrich the gene modules based on their transcriptomic expression and all the genes were assigned into single modules, and one gene cannot be in two or more nodules. Therefore, genes related to migration were divided into different modules due to the different expressed patterns, such as *RELN* involved in M10, *NR4A2* in M16 and *DABI* in M19 (Response Figure 1).

Following the suggestions by the reviewer, we checked the spatial transcriptomic pattern of common genes for migration in the neural development, such as *DCX*. But these common genes were not related to any specific modules because it was highly expressed in all the region (Response Figure 7). From this analysis, we could categorize

genes regulating neuron migration in each region, which are different if they belong to distinctive modules.

Response Figure 7/new Figure S5B

Action taken:

- 1) Following the reviewer's suggestion, we checked spatial transcriptomic patterns of genes related to different modules, such *RELN* in M10, *NR4A2* in M16 and *DAB1* in M19. We found that patterns of these genes were consistent to their modules (Response Figure 7/new Figure S5B) and the common migrating gene *DCX* expressed in all regions could not be concluded into any modules, which show that the gene modules are spatial-specific.
- 2) We added the spatial transcriptomic patterns of *DCX*, *RELN*, *NR4A2* and *DAB1* in the new Figure S5B to make this part more readable.

2. Page 7, Lines 195-197: *The manuscript states that 'Moreover, genes related to glial differentiation, such as TTYH1, GFAP and FOXJ1, were expressed at high levels in Progenitor 6 and Progenitor 8, suggesting the glial cell fate of these progenitors (Figure 2D, E).' The glial cell fate from VZ lineage isn't shown in the Figure 2E. Can you also elaborate on the timeline of VZ progenitors differentiate into each glial subtype in the human developing cerebellum?*

Response: We are grateful for the reviewer's suggestion and completely agree that addressing the differentiate sequences of VZ progenitors differentiate into each glial subtype in the human developing cerebellum would be a good way to improve our study. Therefore, we employed the Velocity Plot to show the differentiate fates of Progenitor 5, 6 and 8 using the same layout with Figure 2E but found that we could not clearly

figure out the cell fates of these glial progenitors (Response Figure 8). One reason could be that the UMAP layout was not the good visualization for trajectory.

Then, we employed URD plot to show the differentiation of VZ progenitors into each glial subtype. As the Response Figure 8 (right panel) shown, Progenitor 5 is the proliferating stem cell in VZ, the remaining cells were distributed along pseudo-temporally ordered paths from progenitors to OPCs and oligodendrocytes and soon afterwards, to astrocytes and Bergmann cells, which is consistent with the glial development in mice (Carter et al.,2018,Vladoiu et al.,2019).

Response Figure 8/ new Figure S6G, S6H

Action taken:

- 1) We analyzed the glial progenitors' cell fates with URD to identify how glial progenitor differentiate into each glial subtypes as Response Figure 8 shown. We found that Progenitor 5 is the proliferating progenitor group and then would be differentiate into Progenitor 6 and 8. Then Progenitor 6 would become astrocytes or Bergmann cells while Progenitor 8 would divide into OPC and Oligodendrocytes. We added new Figure S6G, S6H and revised the manuscript accordingly (Page 7, Line 201-203).

3. Pages 6-7: Does the RL lineage differentiation follow the sequence of eCN, granule cell, and UBCs in order? Is this timeline different between humans and mice?

Response: We thank the reviewer's suggestions and completely agree that the RL lineage differentiation sequence is very important for understanding of cerebellum development. However, in Figure 2G, we built a regulatory hierarchical dendrogram by RL lineages in order to show how the TF regulons drive RL progenitor differentiating into eCN, granule cell and UBCs. The hierarchical plots depicted that eCN showed more different features with granule cells/UBCs in the transcriptomic level but not in the developing time.

Following the reviewer's suggestion, we analyzed the proportions of eCNs, granule cells and UBCs from GW12 to GW27 based on our datasets. We found that eCNs

maintained a stable proportion along the development and granule cells were getting a larger group as the time pass by, while UBCs increased initially and decreased afterwards (Response Figure 9). Based on the proportions of eCNs, granule cells and UBCs, we believed that the RL lineage differentiation follows the order of eCN, granule cell, and then UBCs.

The timeline of mice cerebellum development has shown the RL lineage differentiation follows the sequence of eCNs, granule cells and UBCs in order according to the reported work (Response Figure 10)(Carter *et al.*,2018,Vladoiu *et al.*,2019). The RL lineage differentiation sequence seemed to be conserved between human and mice.

Response Figure 9

Response Figure 10(Carter *et al.*,2018)

Action taken:

- 1) In order to analyze the RL lineage differentiation sequence, we calculated the proportions of eCNs, granule cells and UBCs from GW12 to GW27 in the embryonic developing cerebellum (Response Figure 9). Based on the proportions of eCNs, granule cells and UBCs, we think that the RL lineage differentiation follow the sequence of eCN, granule cell, and UBCs in order, which is consistent with mice (Response Figure 10)(Carter *et al.*,2018,Vladoiu *et al.*,2019). We added one

sentence to the discussion as “the overall cell type emerging sequence is conserved” (Page 16, Line 445-446)

4. Pages 9-10: The discovery of different Purkinje subtypes is potentially significant. However, given the limited information, understanding the developmental differences of these subtypes is challenging. Do Purkinje cells exhibit varying maturation statuses in different subtypes within the developing cerebellum?

Response: We are grateful for the reviewer’s comments. Following the reviewer’s suggestion, we checked the maturation of different subtypes of Purkinje cells using monocle3 in revised new Figure 3E,3F. We found that, C8 and C9 were likely the precursors of Purkinje cells with high expression of *NFIA/NFIB* and *SOX2* shown in the new Figure 3D-F (Response Figure 11,12). Then we analyzed the location of these cells and found that they located in the VZ region shown in the Response Figure 12/ new Figure 3E, which supported the hypothesis that C8 and C9 were precursor groups. C1 seemed to be the most mature subtype with high expression with *RORB* and *LMO4* shown in the new Figure 3D-F (Response Figure 11,12). C1 seemed to be located in the furthest distance from VZ region which implied that this group of Purkinje cells might migrate for long distance. Understanding of the variation of Purkinje cells development would be a good way to improve our study. We revised the figures and text accordingly (Page 10, Lines 263-266).

Response Figure 11/ new Figure 3F

Response Figure 12/ new Figure 3E

Action taken:

- 1) In order to check the varying maturation statuses in different subtype of Purkinje cells as the reviewer’s suggestion, we performed trajectories analysis on scRNA and spatial transcriptomic dataset, respectively.
- 2) We added the trajectory analysis as new Figure 3E and 3F. Also, we revised the figures and text accordingly (Page 10, Lines 263-266).

5. Pages 10-11: The development and migration of granule cells are perhaps the most exciting aspects of this study, especially the regulation of EGL migration to form IGL. The interactions of these migrating granule cells with the Purkinje Layer and Bergmann glial cells are still unclear. Do the multi-omic datasets reveal any new insights into these interactions or regulations?

Response: We are grateful for the reviewer’s constructive comments and suggestion. We added the iTALK analysis to dig out the ligand-receptor interaction between Purkinje cells/Bergmann cells and granule cells as follow, separately. We found that both Purkinje cells and Bergmann cells would secrete growth factor PTN to promote granule cells migrating (Tang et al.,2019,Qin et al.,2017). We revised the figures (Response Figure 13, revised Figure S10E-G) and text accordingly (Page 11, Lines 305-310).

Response Figure 13/ new Figure S10E-G

Action taken:

- 1) Following the reviewer's suggestion, we performed iTalk analysis between Purkinje cells/Bergmann cells and granule cells to find out the ligand-receptor interaction. Then GO terms analysis was performed for these ligand-receptors, we both found that Purkinje cells and Bergmann cells would secrete PTN to drive granule cell migration.
- 2) We added Response Figure 7 into Figure S10 and revised the manuscript (Page 11, Lines 305-310).

6. Pages 13-14: The analysis of cell types and disease-associated SNPs in coding and non-coding regions is impressive. Are there any differences between these two datasets? Do some genes both mutated in coding and non-coding regions?

Response: We thank the reviewer for the insightful comments. We conducted enrichment analysis for scRNA-seq data for the coding region shown in the Figure 6A (left part) and Figure S12A and snATAC-seq data for the non-coding region shown in the Figure 6A (right part) and Figure 6B. We revised the descriptions in the text accordingly to make it more readable (Page 14, Lines 388-395).

We are grateful for the reviewer's suggestions and completely agree that the analysis of whether the SNPs respectively located in coding and non-coding regions link to same genes and the same disease phenotype would strengthen the depth of our findings. By further analysis, we observed SNPs both located in coding and non-coding regions link to the same gene. We found that 264 genes with coding and non-coding SNPs at the same time in our datasets, and 46.59% of them associated with different disease phenotype, and we added the Table S15 accordingly.

Action taken:

- 1) We revised the descriptions of the enrichment analysis conducted separately on scRNA-seq and snATAC-seq in the text (Page 14, Lines 388-391).
- 2) Following the reviewer's suggestion, we analyzed the genes with SNPs in both coding and non-coding regions at the same time. We observed 1619 genes in Figure 6A, and found that 1355 genes (83.7%) with SNPs either in coding region or non-coding region while 264 genes (16.3%) with SNPs both in coding and non-coding region. Among these 264 genes, we found 46.59% of them associated with different disease phenotype. We presented information of these 264 genes in the new Table S15 and revised the manuscript accordingly (Page 14, Lines 391-395).

Minor points:

1. Page 2, Line 45-48: *'we combined single-cell transcriptomics, spatial transcriptomics and chromatin accessibility states ...'*, the authors used the description of *'spatial transcriptomics and chromatin accessibility states'* misled the concept that you use spatial ATAC-seq data, please correct the description.

Response: Following the reviewer's suggestion, we revised the description in the text accordingly (Page 2, Line 45-48).

2. Page 2, Line 48-49: *'Our multiomic data revealed that combinations of transcription factors at CREs play roles in ...'*. Please correct the word *'at'*.

Response: We thank the reviewer for pointing this out, and we corrected it in text (Page 2, Line 48-49).

3. In Figure 1D, the gene name of granule cell group was missing.

Response: We thank the reviewer for pointing this out, and we corrected it in Figure 1D.

4. In Figure 2G, some gene names are unclear.

Response: We thank the reviewer for pointing this out, and we corrected it in Figure 2G.

5. Page 7, Line 196: *FOXJ1* seems not in Figure 2D,E.

Response: We thank the reviewer for pointing this out and apologize for the not depicting the FOXJ1 pattern in the heatmap from Figure 2D and revised the text in the manuscript to make it more readable (Page 7, Lines 199-200).

References.

1. Carter, R.A., Bihannic, L., Rosencrance, C., Hadley, J.L., Tong, Y., Phoenix, T.N., Natarajan, S., Easton, J., Northcott, P.A., and Gawad, C. (2018). A Single-Cell Transcriptional Atlas of the Developing Murine Cerebellum. *Curr Biol* 28, 2910-2920 e2912. 10.1016/j.cub.2018.07.062.
2. Vladoiu, M.C., El-Hamamy, I., Donovan, L.K., Farooq, H., Holgado, B.L., Sundaravadanam, Y., Ramaswamy, V., Hendrikse, L.D., Kumar, S., Mack, S.C., et al. (2019). Childhood cerebellar tumours mirror conserved fetal transcriptional programs. *Nature* 572, 67-73. 10.1038/s41586-019-1158-7.
3. Tang, C.Y., Wang, M., Wang, P.J., Wang, L., Wu, Q.F., and Guo, W.X. (2019). Neural Stem Cells Behave as a Functional Niche for the Maturation of Newborn Neurons through the Secretion of PTN. *Neuron* 101, 32-+. 10.1016/j.neuron.2018.10.051.

4. Qin, E.Y., Cooper, D.D., Abbott, K.L., Lennon, J., Nagaraja, S., Mackay, A., Jones, C., Vogel, H., Jackson, P.K., and Monje, M. (2017). Neural Precursor-Derived Pleiotrophin Mediates Subventricular Zone Invasion by Glioma. *Cell* *170*, 845-859 e819. [10.1016/j.cell.2017.07.016](https://doi.org/10.1016/j.cell.2017.07.016).

Reviewer #1 (Remarks to the Author):

The authors have addressed all my questions.

Reviewer #2 (Remarks to the Author):

The authors have addressed all my previous comments. I believe this revised version is ready to be published.

REVIEWERS' COMMENTS

Reviewer #1 (Remarks to the Author):

The authors have addressed all my questions.

Response: We appreciate the time and effort that the reviewer dedicated to providing feedback on our manuscript and are grateful for the insightful comments on and valuable improvements to our paper.

Reviewer #2 (Remarks to the Author):

The authors have addressed all my previous comments. I believe this revised version is ready to be published.

Response: We would like to thank the reviewer for taking the necessary time and effort to review the manuscript. We sincerely appreciate all your valuable comments and suggestions, which helped us in improving the quality of the manuscript.